# Embedding oxophilic rare-earth single atom in platinum nanoclusters for efficient hydrogen electro-oxidation

Xiaoning Wang[1], Yanfu Tong[1], Wenting Feng[1], Pengyun Liu[1], Xuejin Li[1], Yongpeng Cui[1], Tonghui Cai[1], Lianming Zhao[1], Qingzhong Xue[1], Zifeng Yan[1], Xun Yuan [2] ✉ & Wei Xing [1] ✉

Designing Pt-based electrocatalysts with high catalytic activity and CO tolerance is challenging but extremely desirable for alkaline hydrogen oxidation reaction. Herein we report the design of a series of single-atom lanthanide (La, Ce, Pr, Nd, and Lu)-embedded ultrasmall Pt nanoclusters for efficient alkaline hydrogen electro-oxidation catalysis based on vapor filling and spatially confined reduction/growth of metal species. Mechanism studies reveal that oxophilic single-atom lanthanide species in Pt nanoclusters can serve as the Lewis acid site for selective $OH^-$ adsorption and regulate the binding strength of intermediates on Pt sites, which promotes the kinetics of hydrogen oxidation and CO oxidation by accelerating the combination of $OH^-$ and *H/*CO in kinetics and thermodynamics, endowing the electrocatalyst with up to 14.3-times higher mass activity than commercial Pt/C and enhanced CO tolerance. This work may shed light on the design of metal nanocluster-based electrocatalysts for energy conversion.

Hydrogen fuel cells as green yet high-efficiency energy suppliers are very attractive in contributing to fulfilling carbon neutrality, which is greatly benefited from the deployment of precious Pt electrocatalysts performing best in catalyzing anodic hydrogen oxidation reaction (HOR) due to the optimal adsorption/desorption energy toward hydrogen intermediates[1–3]. However, the large-scale implementation of conventional Pt/C electrocatalysts in alkaline HOR catalysis is beset by ultralow catalytic efficiency (e.g., two orders of magnitude decline), extortionate Pt dosage (e.g., ten times higher than that in acidic media), and poor durability in terms of weak CO tolerance and feeble stability[4–16]. Therefore, designing high-performance Pt electrocatalysts with low Pt dosage and high durability is extremely desirable for alkaline HOR, which should be the stone of killing two birds (i.e., low abundance of Pt element and unsatisfactory alkaline HOR performance).

By deciphering the alkaline HOR process[2,17–19], it is found that the mutual competition in the adsorption-desorption between reaction intermediates (e.g., *H and *OH) and CO on the surface-active sites of electrocatalysts is the key reason for the low catalytic activity and poor CO tolerance[20–22]. Although previous researches have improved the alkaline HOR performance of Pt electrocatalysts by decreasing the size or engineering the composition/structure with other metals to conserve the Pt dosage, and regulating the *d*-band center to adjust the adsorption-desorption behavior of *H, *OH, and CO species, they failed to fundamentally solve the mutual competition issues of such reactant species on the active sites of Pt electrocatalysts[11,23]. To radically address such issues, an ideal alkaline HOR electrocatalyst (e.g., platinum group metal (PGM)-based and Ni-based) should feature the following attributes: (1) in addition to the main active sites, another type of active site with selective adsorption-desorption behavior for specific reactant species should be deployed on the electrocatalyst surface to essentially avoid the mutual adsorption-desorption competition between reactant species[5,24–26]; (2) the dissimilar active sites should be well distributed in the electrocatalyst at the atomic level and have

[1]State Key Laboratory of Heavy Oil Processing, School of Materials Science and Engineering, China University of Petroleum, Qingdao 266580, PR China. [2]School of Materials Science and Engineering, Qingdao University of Science and Technology, Qingdao 266042, PR China. ✉e-mail: yuanxun@qust.edu.cn; xingwei@upc.edu.cn

synergistic effects with the main active sites to promote the interfacial HOR kinetics and resist CO poisoning via regulating the *d*-band centers, and concurrently strengthen the structural stability of the electrocatalyst[7,8,17,27,28]; (3) the electrocatalyst should have an ultrasmall size to maximumly uplift its atomic utilization[29–32]. However, such a Pt-based electrocatalyst that combines all three is not yet developed in the field.

Inspired by the conceptual design of high-performance metal nanoclusters (NCs, ≤3 nm)-based nanocatalysts with dissimilar metallic single-atom insertion[33,34], we hypothesized that if we can insert highly oxophilic lanthanide single-atom into ultrasmall Pt NCs ($Ln_1Pt$ NCs for short), we may develop a high-performance electrocatalyst with both excellent catalytic activity and superior durability for alkaline HOR catalysis based on the following considerations: (1) the highly oxophilic $Ln_1$ can serve as Lewis acid to specifically absorb OH species, while the Pt NCs could selectively absorb *H species, which prevent the mutual absorption-desorption competition issue aforementioned; (2) the Ln elements with oxophilicity and atomic-level dispersion in Pt NCs could not only adjust the *d*-band center to regulate the adsorption-desorption behavior of *H and CO, but also allow the fast electron transfer between dissimilar atoms as well as subsequent reaction between *OH and *H/*CO, which accelerates the HOR kinetics, ensures its structural stability through unique architecture protection, and guarantees excellent CO tolerance via prompting *CO oxidation; (3) both the single-atom $Ln_1$ and the ultrasmall size of NCs efficiently enhance atomic utilization. However, the synthesis of such $Ln_1Pt$ NCs is very challenging using conventional wet chemistry methods because the large difference in the standard reduction potentials between Pt (1.188 V for $Pt^{2+}/Pt$) and Ln (e.g., −2.522 V for $La^{3+}/La$, and −2.483 V for $Ce^{3+}/Ce$) usually brings about metallic phase segregation during the synthesis[35], and the low reduction potentials of Ln metals are also far beyond the stable range of water[36]. In addition, the formation of ultrasmall $Ln_1Pt$ NCs is also hardly realized under harsh conditions (e.g., high temperature >1000 °C and strong reducing agents) due to thermodynamic incompatibility between dissimilar metals, let alone to simultaneously avoid the possible low electrochemical active surface area (ECSA), and atomic size mismatch-caused structure collapse of $Ln_1Pt$ NCs (e.g., atomic radiuses are 1.83 Å for Pt and 2.74 Å for La)[37–40]. Taken together, controllable synthesis of ultrasmall $Ln_1Pt$ NCs for alkaline HOR catalysis may rely on the methodological innovation, which makes up part of our motivation of this study.

Herein we report the synthesis of a series of ultrasmall $Ln_1Pt$ NCs with the insertion of highly oxophilic La, Ce, Pr, Nd, and Lu single-atom via vapor filling and spatially confined reduction of metal species inside mesoporous yet hollow carbon spheres-based nanoreactors, achieving high HOR catalytic performance in alkaline media. The key of this synthetic strategy is the atomic level mixing via vapor capillary filling as well as the efficient spatial confinement of metal species using mesoporous channels of carbon nanoreactors at a sublimation temperature, resulting in the formation of ultrasmall $Ln_1Pt$ NCs inside hollow carbon sphere ($Ln_1Pt$@HCS for short, see Fig. 1a). Both experimental and theoretical results reveal that the as-developed $Ln_1Pt$ NCs have reduced work function (Φ) and coordination numbers (CN) due to the pronounced electron transfer/conduction nature of Ln, which enables $Ln_1$ to serve as a Lewis acid site for selective OH⁻ adsorption and regulates binding strength of intermediates on Pt sites, promoting the HOR kinetics and CO tolerance. As a result, the $Ln_1Pt$@HCS yields remarkable electrocatalytic activity towards alkaline HOR (e.g., $La_1Pt$@HCS has a mass activity of 14.3-times higher than commercial Pt/C), and simultaneously achieves enhanced durability in terms of anti-CO poisoning capability and structural stability. This study is interesting because it provides a paradigm in the design of high-performance $Ln_1Pt$ NCs-based electrocatalyst for alkaline HOR catalysis, demonstrating the synergistic power of the single atom and ultrasmall metal NCs in enhancing the catalytic activity, and deepening the fundamental understanding on the Ln/Pt alloy-directed alkaline HOR electrocatalysis.

## Results

### Material synthesis and characterization

The prerequisite of ultrasmall $Ln_1Pt$ NCs synthesis is the acquisition of hollow carbon sphere (HCS) as a nanoreactor. As illustrated in Fig. 1a, the HCS was fabricated by carbonizing core-shell resorcinol-formaldehyde resin@silica nanospheres (prepared by a hard-template method) followed by removing the silica core (see "Methods" for details)[41]. The obtained HCS is a particle of ~250 nm containing a regular spherical cavity surrounded by a uniform shell of ~50 nm, as evidenced by both scanning electron microscopy (SEM, Supplementary Fig. 1) and transmission electron microscopy analysis (TEM, Fig. 1b). Such an HCS can be regarded as a well-defined nanoreactor upon silica removal, consisting of mesoporous shells and pore channels with radial alignment. Moreover, the porous HCS material has a large Brunauer–Emmett–Teller surface area of over 1505.8 $m^2 g^{-1}$ and a high pore volume of 2.2 $cm^3 g^{-1}$ (Supplementary Fig. 2). Of note, the HCS also possesses tailored pore channels of 5.1 nm calculated by the density function theory (DFT) method (Fig. 1c), which would easily accommodate and confine metal species while ensuring adequate mass transport of electrolyte and gas.

With HCS nanoreactors in our hand, ultrasmall $Ln_1Pt$ NCs could be synthesized inside HCS via vapor filling of Ln and Pt precursors into mesoporous channels of HCS followed by spatially confined in-situ reduction/growth[16]. Taking the synthesis of $La_1$-doped $La_1Pt$@HCS as an example, gas-migrated $Pt(acac)_2$ and $La(acac)_3$ precursors were injected into the pore channels in a vacuum-sealed ampoule at a preset sublimation temperature. After successive in-situ pyrolysis, the metal precursors decomposed and reassembled into $La_1Pt$ NCs under the confinement of pore channels (Fig. 1a). The TEM and SEM images of $La_1Pt$@HCS (Fig. 1d–f) unveil that abundant $La_1Pt$ NCs with an average size of ~2.6 nm are well distributed inside the HCS and probably confined in the pore channels rather than on the surface of HCS. The metallic composition of $La_1Pt$@HCS was comprehensively analyzed by the high-angle annular dark-field scanning TEM (HAADF-STEM) and elemental mapping. As shown in Fig. 1g, the low-magnification HAADF-STEM image and line-scanning profile disclose that the embedded NCs consist of bimetallic alloys including Pt and La, while the aberration-corrected (AC)-HAADF-STEM image of $La_1Pt$@HCS further reveals that La atoms are atomically dispersed (Fig. 1h) in Pt crystals, and the inserted selected area electron diffraction (SAED) pattern depicts its typical Pt crystal (inset of Fig. 1h). Along lines 1 and 2 in Fig. 1h, $La_1Pt$@HCS delivers an obvious intensity variation of Pt and La atoms in Z contrast, as observed by the intensity profiles in Fig. 1i. Energy-dispersive X-ray (EDX) elemental mappings in Fig. 1j show even spatial distribution of Pt, La, C, and O elements. In addition, the total Pt to La ratio of $La_1Pt$@HCS was tested to be 8.8:1 based on inductively coupled plasma optical emission spectroscopy (ICP-OES) (Supplementary Table 1).

The above-mentioned results certify the formation of single-atom $La_1$ which is well embedded inside the ultrasmall Pt NCs in the HCS, reflecting the successful design of the synthetic strategy. Furthermore, large-scale fabrication of $La_1Pt$@HCS could be easily realized using this approach (Supplementary Fig. 3). It should be mentioned that the mesoporous channels of HCS are crucial to the formation of such ultrasmall NCs because of their effective size confinement for metal species, which is supported by two control experiments: (1) using XC-72 carbon material (Vulcan®, BET surface area: 211.5 $m^2 g^{-1}$) to replace HCS but without introducing La species, resulting in the generation of large-sized Pt nanoparticles (~12.8 nm) on the XC-72 surface

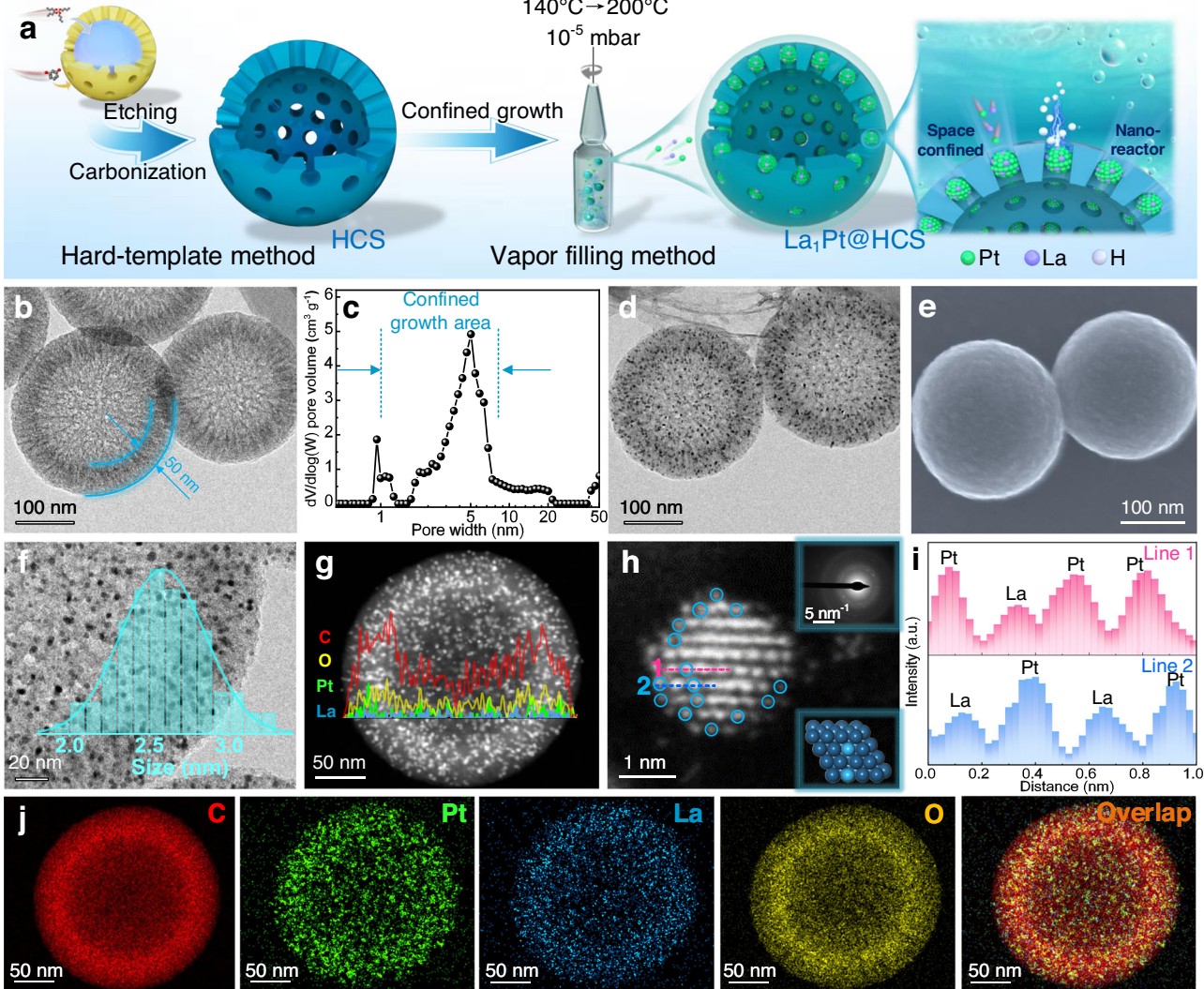

**Fig. 1 | Synthesis, morphological and structural characterization of La1Pt@HCS electrocatalyst. a** Schematic illustration on the synthesis of La1Pt@HCS. **b** TEM image of HCS. **c** Pore size distribution of HCS estimated by DFT method. **d** TEM image, **e** SEM image, **f** Enlarged TEM image (inset: size distribution histogram), **g** HAADF-STEM image (inset: line scan of elemental profiles) of La1Pt@HCS.

**h** Atomic-resolution HAADF-STEM image of La1Pt@HCS (inset: SAED pattern image and the atomic structure model). **i** Intensity profiles along lines 1 and 2 indicated in (**h**). **j** STEM-EDX mapping images of La1Pt@HCS showing distributions of C (red), Pt (green), La (blue), O (yellow), and merged elements.

(Supplementary Fig. 4); (2) using commercially available porous activated carbon (Kuraray® YP-50, PAC for short; BET surface area: 1568.6 $m^2$ $g^{-1}$) to replace HCS, leading to the successful production of La1Pt NCs (~3.1 nm) inside PAC (Supplementary Fig. 5). In addition, one salient advantage of this newly developed strategy is the universal applicability, and other Ln1Pt@HCS electrocatalysts (here Ln = Ce, Pr, Nd, and Lu) can also be obtained by this method if introducing corresponding rare-earth precursors with unique 4f shell electron orbitals and high oxophilicity properties (Supplementary Figs. 6–11).

The embedding of highly oxophilic La1 single-atom in Pt NCs exerts significant influences on the work function ($\Phi$), d-band center, and coordination numbers (CN) of Pt NCs, which are key factors affecting the HOR catalytic activity. To elucidate the impact of La1 embedding, we used La1Pt@HCS as an example to compare with the same-sized Pt NCs without La1 embedment inside HCS (Pt@HCS, and its characterization is shown in Supplementary Fig. 12). As shown in Fig. 2a, both La1Pt@HCS and Pt@HCS share the same XRD patterns consisting of the diffraction peaks of face-centered cubic (fcc) Pt crystal (JCPDS-PDF#87-0646) and the (002) graphite carbon peak belonging to HCS, which suggests that the La1 embedment with a low doping dosage (~0.98 wt.%) has a neglectable influence on the crystal

structure of Pt NCs. On the other hand, the oxidation state of Pt 4f X-ray photoelectron spectra (XPS, Fig. 2b) of Pt NCs has remarkable variation upon La1 embedment, and the proportion of $Pt^{2+}$ is increased from 43.8% in Pt@HCS to 56.2% in La1Pt@HCS due to slight ionization (Fig. 2b)[29,42,43]. In parallel, the doublet separation energy of La1Pt@HCS shown in the La 3d XPS (Fig. 2c) is reduced to 3.1 eV compared to 3.9 eV for La(OH)3 and 4.6 eV for La2O3. Of note, the decreased splitting energy of La1 in La1Pt@HCS signifies the reduction of its electron cloud density because the splitting energy is originated from spin-orbit coupling, resulting in the enhanced binding energy of La1 with reaction intermediates[39,44].

It is well documented that the lower $\Phi$ means a higher d-band center and electron-delocalization capability, which favors the escape of electrons to intervene in the HOR process[45]. Figure 2d displays the $\Phi$ extracted by ultraviolet photoelectron spectroscopy (UPS), where the secondary electron cutoff energy ($E_{cutoff}$) delivers a decrease from 17.62 eV for Pt@HCS to 16.77 eV for La1Pt@HCS. Correspondingly, the $\Phi$ decreases from 4.45 eV for Pt@HCS to 3.96 eV for La1Pt@HCS. Meanwhile, the valence band maximum position ($E_{VBM}$) of La1Pt@HCS shifts negatively by 0.13 eV compared to that of Pt@HCS. The valence band shifts moderately to the Fermi level upon La1 embedment,

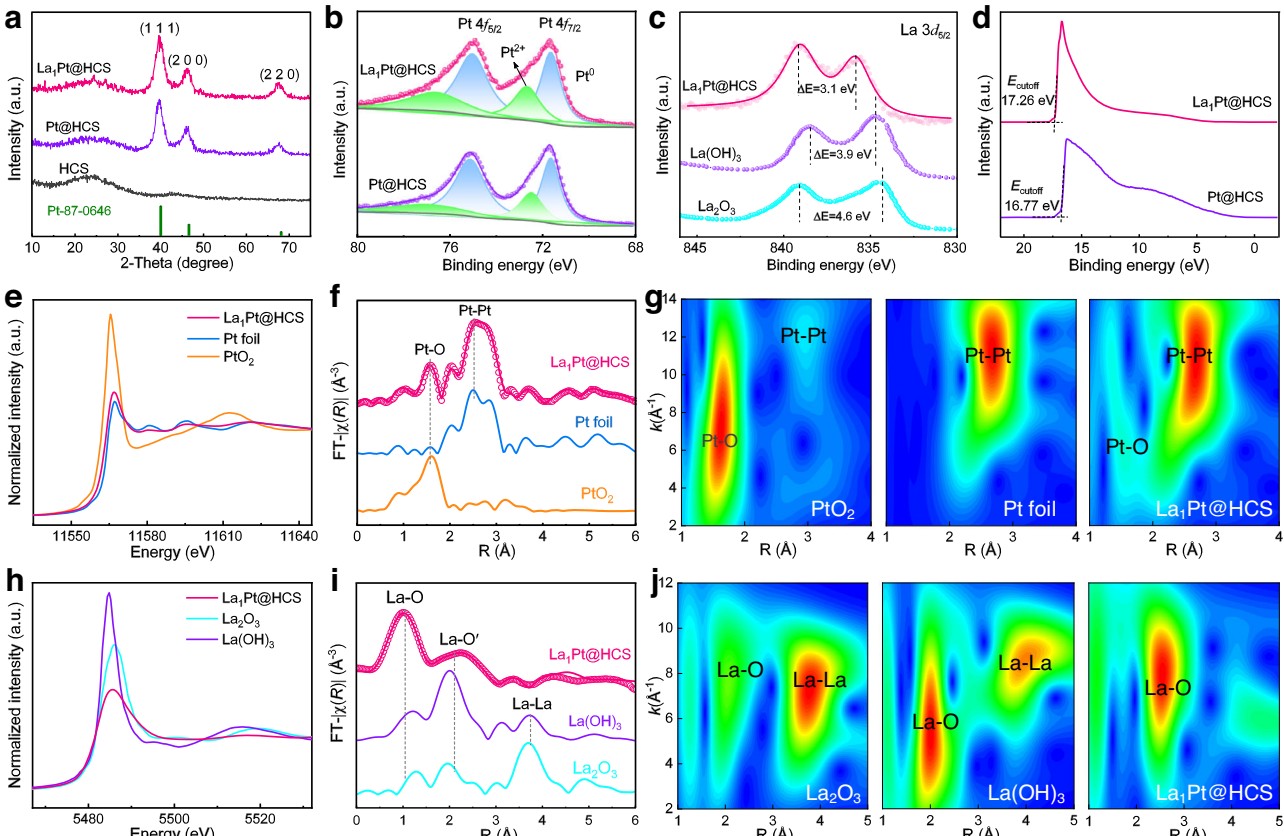

**Fig. 2 | Structural characterizations. a** XRD patterns of La$_1$Pt@HCS, Pt@HCS, and HCS. **b** Pt 4$f$ XPS spectra of La$_1$Pt@HCS and Pt@HCS. **c** La 3$d_{5/2}$ XPS spectra of La$_1$Pt@HCS, La(OH)$_3$, and La$_2$O$_3$. **d** UPS spectra of La$_1$Pt@HCS and Pt@HCS. **e** Normalized XANES at Pt L$_3$-edge for La$_1$Pt@HCS with Pt metal foil and PtO$_2$ as references. **f** FT-EXAFS curves at Pt L$_3$-edge, where the curve is experimental data and the circle is the best fit for La$_1$Pt@HCS. **g** WT-EXAFS signals at Pt L-edge for La$_1$Pt@HCS and the references. **h** Normalized XANES at La L$_3$-edge for La$_1$Pt@HCS with La(OH)$_3$ and La$_2$O$_3$ as references. **i** FT-EXAFS curves at La L$_3$-edge, where the curve is experimental data and the circle is the best fit for La$_1$Pt@HCS. **j** WT-EXAFS signals at La L$_3$-edge for La$_1$Pt@HCS and the references.

suggesting that the $d$-band center of La$_1$Pt@HCS upshifts compared to that of Pt@HCS[46]. These results corroborate a properly enhanced OH⁻ adsorption capacity of La$_1$Pt@HCS, which is beneficial to accelerate the *H removal and interfacial water formation.

We also assessed the effect of La$_1$ embedment on the atomic coordination and structural signature of La$_1$Pt@HCS by X-ray absorption near-edge structure (XANES) and extended X-ray absorption fine structure (EXAFS) measurements. Figure 2e presents the Pt L$_3$-edge XANES spectra for La$_1$Pt@HCS and the references (Pt foil and PtO$_2$). The white line intensity peak of La$_1$Pt@HCS is located between those of Pt foil (Pt$^0$) and PtO$_2$ (Pt$^{4+}$), suggesting that the average oxidation state of Pt is the partial positive charge state due to the electron transfer from Pt to La−O/O species[47,48], consistent with the XPS analysis (Fig. 2b). Of note, such an increased oxidation state and $d$ orbital occupation correlate with the enhanced OH⁻ adsorption capacity under alkaline conditions, which favors alkaline HOR kinetics[12]. Moreover, the Pt L-edge EXAFS spectra were obtained by Fourier transformation (FT) to elucidate bonding information (Fig. 2f). As shown, a dominant peak with the feature of the Pt foil and a second-intensive peak similar to that of the PtO$_2$ are observed for La$_1$Pt@HCS, which could be ascribed to Pt−Pt and Pt−O contributions, respectively. The FT-EXAFS curves at Pt L$_3$-edge were fitted in $R$ space (Supplementary Fig. 13), and those fitting results are summarized in Supplementary Table 2. The lower Pt-Pt average CN (5.1) and longer Pt−Pt bond length (3.05 Å) than those of Pt foils (12, 2.77 Å) reconfirm the tendency to generate weak $d$-orbital interactions and an upshift of the $d$-band center, resulting in a promoted OH⁻ adsorption capacity. In addition, wavelet transform (WT) analysis was also conducted to

obtain more intuitive bonding information (Fig. 2g), and the result discloses that the La$_1$Pt@HCS shows both the intensity maxima of Pt foil and PtO$_2$, which is consistent with the EXAFS result.

There may be a concern about whether the embedded La species in Pt NCs are single-atom La$_1$ or aggregate of >2 La atoms, which could also be proven by XANES and EXAFS. As revealed by the La L$_3$-edge XANES spectra (Fig. 2h), the white line intensities gradually decline in the order of La(OH)$_3$, La$_2$O$_3$, and La$_1$Pt@HCS, suggesting a lower La coordination number in La$_1$Pt@HCS, which is consistent with the low coordination situation of atomically dispersed La$_1$ within the Pt NCs[40]. Figure 2i shows the FT-EXAFS for La L$_3$-edge of La$_1$Pt@HCS, La(OH)$_3$, and La$_2$O$_3$ references. The $k^3$-weight EXAFS spectrum at La L$_3$-edge is obtained to identify the coordination environment in $R$ space (Supplementary Fig. 14), and those fitting results are summarized in Supplementary Table 3. The FT-EXAFS curve of La$_1$Pt@HCS (Fig. 2i) manifests two main peaks between 1.0 and 2.5 Å, related to La−O scattering similar to that of La(OH)$_3$ and La$_2$O$_3$. The La−O bond might be due to the oxidation of the single atom alloy catalyst during ex situ tests[13,34]. There is no La−La contributing peak at around 4.0 Å in La$_1$Pt@HCS, validating that the La species exists in an isolated monoatomic state without long-range coordination to other La centers. This finding is consistent with the observation of La single atoms embedded within Pt NCs in the AC-HAADF-STEM image (Fig. 1h). WT-EXAFS analysis (Fig. 2j) at La L$_3$-edge manifests one intensity maximum at ~2.5 Å related to the La−O path. It is probably reasonable to assign the second intensity peak at ~4.5 Å in the WT-EXAFS analysis to the Pt−La path. Taken together, we may conclude that single La atom is coordinated to Pt, giving rise to the Pt−La configuration. More

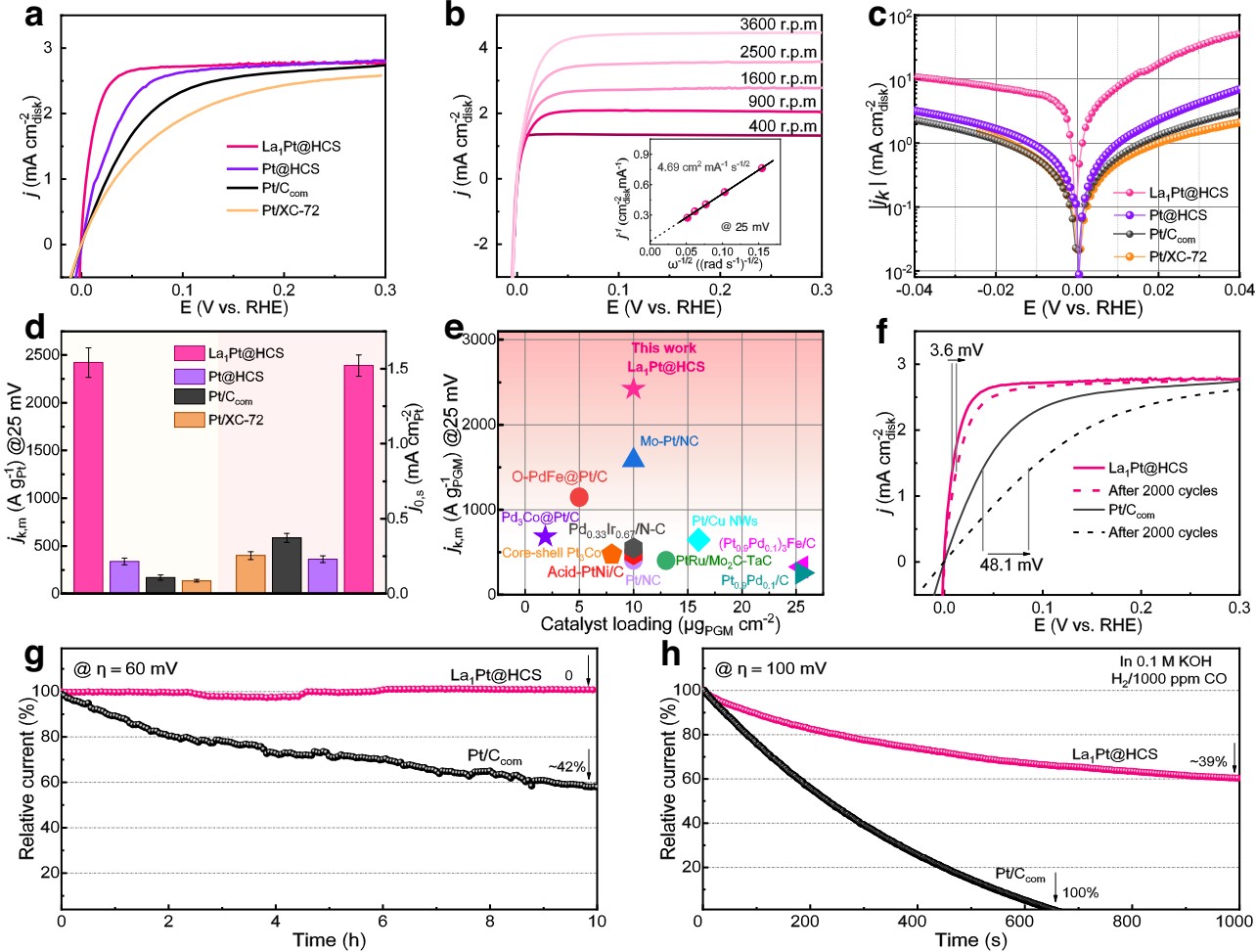

**Fig. 3 | Electrocatalytic HOR performance. a** HOR polarization curves on La$_1$Pt@HCS, Pt@HCS, Pt/C$_{com}$, and Pt/XC-72 in H$_2$-saturated 0.1 M KOH with a scan rate of 5 mV s$^{-1}$ at a rotating rate of 1600 rpm. **b** HOR polarization curves of La$_1$Pt@HCS at varied rotating speeds (inset: corresponding Koutechy-Levich plots). **c** Tafel plots of the kinetic current densities. **d** Normalized mass activity ($j_{k,m}$) at an overpotential of 25 mV (vs. RHE) and specific activity ($j_{0,s}$). The error bars (standard deviations) in (**d**) were calculated from at least three independent testing results. **e** Comparison of the $j_{k,m}$ of La$_1$Pt@HCS at 25 mV (vs. RHE) with that of other recently reported PGM-based electrocatalysts. **f** HOR polarization curves of La$_1$Pt@HCS and Pt/C$_{com}$ before and after ADT. **g** Chronoamperometry ($j$–$t$) response of La$_1$Pt@HCS and Pt/C$_{com}$ catalysts in an H$_2$-saturated 0.1 M KOH at 60 mV (vs. RHE). **h** Chronoamperometry ($j$–$t$) response of La$_1$Pt@HCS and Pt/C$_{com}$ catalysts in CO (1000 ppm)/H$_2$-saturated 0.1 M KOH at 100 mV (vs. RHE).

importantly, the La$_1$ embedment in Pt NCs could theoretically favor alkaline HOR catalysis because the electrons of La$_1$Pt@HCS could flow from La$_1$ into the Pt species, leaving the positively ionized donor behind as a Lewis acid site to adsorb hydroxyl species and accelerate the Volmer step of HOR[44]. As known, a typical HOR usually follows the Tafel (H$_2$ + 2 M → 2M-*H)-Volmer (M − *H + OH$^-$ → M + H$_2$O + e$^-$) or Heyrovsky (H$_2$ + OH$^-$ + M → M − *H + H$_2$O + e$^-$)-Volmer pathway (M, catalytic site; *H, adsorbed hydrogen).

## Evaluation of HOR performance

As expected, the as-developed La$_1$Pt@HCS electrocatalyst achieves high catalytic activity for alkaline HOR, which is evidenced by the HOR polarization results. The HOR polarization was tested via a rotating disk electrode method in an H$_2$-saturated 0.1 M KOH electrolyte with Pt@HCS, Pt/XC-72, and benchmark 20% Pt/C (Pt/C$_{com}$) as references. As shown in Fig. 3a, the HOR polarization curve of La$_1$Pt@HCS recorded at a rotating speed of 1600 rpm manifests a sharp current response in an H$_2$-rich environment, which elucidates that the anodic current originates from H$_2$ oxidation considering the negligible current-voltage in N$_2$-saturated 0.1 M KOH (Supplementary Fig. 15). Moreover, the half-wave potential ($E_{1/2}$) of La$_1$Pt@HCS is determined to be 8.4 mV (vs. RHE, all potentials in this work are given versus RHE),

which is much smaller than 25.0 mV for Pt@HCS, 38.8 mV for Pt/C$_{com}$, and 48.5 mV for Pt/XC-72 catalysts, suggesting the best kinetic activity of La$_1$Pt@HCS for alkaline HOR catalysis. Meanwhile, the Pt@HCS delivers a sharper anodic current increment than Pt/XC-72 at the kinetic control regions, implying its better active site dispersion and higher atomic utilization triggered by the confined growth of Pt species.

The H$_2$ mass transport as an important factor for HOR kinetics was also evaluated by analyzing the polarization process of La$_1$Pt@HCS and reference electrocatalysts at different rotating speeds (Fig. 3b and Supplementary Fig. 16). As shown, the increased current along with the elevated rotation rate reveals the H$_2$ mass transport, with a linear relationship between the inverse of the total current and the square root of the rotational speeds (inset in Fig. 3b). The slope of the Koutechy-Levich plot for La$_1$Pt@HCS is determined to be 4.69 cm$^2$ mA$^{-1}$ s$^{-1/2}$ at 25 mV, which is near the theoretical value of 4.87 cm$^2$ mA$^{-1}$ s$^{-1/2}$ for a two-electron HOR process[31]. The kinetic current density ($j_k$) of La$_1$Pt@HCS, Pt@HCS, Pt/C$_{com}$, and Pt/XC-72 can be extracted from the Koutechy−Levich formula to quantitatively estimate their reaction rate and activity (Fig. 3c). After normalizing the mass of Pt on La$_1$Pt@HCS based on ICP-OES, the mass activity ($j_{k,m}$) of La$_1$Pt@HCS at 25 mV reaches 2422.38 A g$^{-1}$ (Fig. 3d), which is 7.2, 14.3, and 18.0 times higher

than that of Pt@HCS ($338.25\,A\,g^{-1}$), Pt/C$_{com}$ ($169.05\,A\,g^{-1}$), and Pt/XC-72 ($134.29\,A\,g^{-1}$), respectively. To the best of our knowledge, the remarkable mass activity (at 25 mV) of La$_1$Pt@HCS outperforms that of most PGM-based electrocatalysts (Fig. 3e).

Moreover, the exchange current density ($j_0$) for La$_1$Pt@HCS is determined to be $20.08\,mA\,cm^{-2}$ calculated by fitting the $j_k$ according to the Butler–Volmer formula, which is higher than that of Pt@HCS ($2.61\,mA\,cm^{-2}$), Pt/C$_{com}$ ($1.55\,mA\,cm^{-2}$), and Pt/XC-72 ($1.40\,mA\,cm^{-2}$). To quantitatively compare their intrinsic HOR activity, we calculated the specific activity normalized with the ECSA ($j_{0,s}$). As the ECSA is determined to be $129.5\,m^2\,g^{-1}$ for La$_1$Pt@HCS, $112.3\,m^2\,g^{-1}$ for Pt@HCS, $42.8\,m^2\,g^{-1}$ for Pt/C$_{com}$, and $55.1\,m^2\,g^{-1}$ for Pt/XC-72 based on CO-stripping measurements (Supplementary Fig. 17), the $j_{0,s}$ of La$_1$Pt@HCS was thus calculated to be $1.55\,mA\,cm^{-2}$, which is 6.7, 4.3, and 6.2 times higher than that of Pt@HCS ($0.23\,mA\,cm^{-2}$), Pt/C$_{com}$ ($0.36\,mA\,cm^{-2}$), and Pt/XC-72 ($0.25\,mA\,cm^{-2}$), respectively. This result also ascertains the positive effect of spatially confined growth on Pt atom utilization/dispersion. The $j_{0,s}$ of some PGM-based electrocatalysts are summarized in Supplementary Table 4, which highlights the superior catalytic performance of La$_1$Pt@HCS.

In addition to high electrocatalytic activity, the as-developed La$_1$Pt@HCS also displays excellent durability in terms of performance stability and anti-CO poisoning capability, which is another prominent feature for advanced electrocatalysts. First, we conducted an accelerated durability test (ADT) by applying continuous potential cycling on La$_1$Pt@HCS and Pt/C$_{com}$ catalysts in H$_2$-saturated 0.1 M KOH. After 2000 cycles, La$_1$Pt@HCS delivers a 3.6 mV increase in $E_{1/2}$, versus 48.1 mV in Pt/C$_{com}$ (Fig. 3f), revealing the high-performance stability of La$_1$Pt@HCS catalyst. Second, the chronoamperometry results further support the excellent performance stability of La$_1$Pt@HCS (Fig. 3g), where La$_1$Pt@HCS shows a current density with almost zero decay after 10 h at 60 mV, which is in stark contrast to the Pt/C$_{com}$ losing ~42% of HOR activity under the same conditions. It is worth noting that the high-performance stability of La$_1$Pt@HCS is derived from their unique HCS architecture-mediated protection for La$_1$Pt NCs, and we compared the surface chemical states and morphology of La$_1$Pt@HCS before and after ADT to clarify this issue. As shown in Supplementary Fig. 18, SEM-EDS images show that the morphology and composition of the spent La$_1$Pt@HCS are largely maintained. The invariant Pt 4$f$ and La 3$d$ XPS spectra of La$_1$Pt@HCS before and after ADT corroborate its good surface chemical stability during the long-term electrochemical process (Supplementary Fig. 19). In addition, the TEM image of the spent La$_1$Pt@HCS reveals that the La$_1$Pt NCs are still space-confined on nanopores without apparent morphological changes (Supplementary Fig. 20). These results elucidate the superior HOR catalytic stability of La$_1$Pt@HCS catalyst, which stems from the confined growth induced by the intrinsic porosity of HCS, effectively inhibiting the agglomeration of Pt species.

In parallel, the La$_1$Pt@HCS also exhibits excellent long-term anti-CO poisoning ability with Pt/C$_{com}$ as a reference. It is known that CO poisoning is an intractable obstacle for anodic PGM-based electrocatalysts (especially Pt) when using fossil fuels for fuel H$_2$ production, and the preferential CO binding on Pt surface could poison the active sites for *H adsorption/desorption. In this study, the anti-CO poisoning ability of La$_1$Pt@HCS was evaluated by a chronoamperometry at 100 mV with 1000 ppm CO impurity. As illustrated in Fig. 3h, the La$_1$Pt@HCS catalyst retains a high catalytic activity (current retention rate: ~61%) in the presence of CO impurity after continuously operating for ≈1000 s, whereas Pt/C$_{com}$ becomes completely inactive (current decay rate: 100%) after continuous HOR catalysis (≈650 s), corroborating the superior anti-CO poisoning capability of La$_1$Pt@HCS comparing to Pt/C$_{com}$. The robust anti-CO poisoning ability of La$_1$Pt@HCS is due to that the Pt-coordinated La$_1$ single atom with strong OH$^-$ adsorption capacity could promote the oxidation of *CO[4]. Taken together, the aforementioned results demonstrate that the La$_1$Pt@HCS can serve as an efficient yet durable electrocatalyst for alkaline HOR.

## Mechanism investigation

We are curious about the atomic/molecular-level catalytic mechanism of the La$_1$Pt@HCS in alkaline HOR, and thus employed DFT simulation to approach this goal. The Pt(111) surface is selected in DFT calculations because the (111) plane possesses the lowest surface energy and the most-abundant surface sites in Pt crystal[33]. Firstly, the model of La$_1$Pt@HCS was constructed by using one La atom to substitute for one Pt atom in Pt(111) (Supplementary Fig. 21) based on the XRD, STEM, and XAS data. Such a La$_1$-embedded configuration has lower formation energy (−4.031 eV) than the La$_1$-deposited one (fcc site: −3.151 eV, hcp site: −3.077 eV; Supplementary Fig. 22), further supporting that the La$_1$ was embedded in Pt NCs[49]. With the model in hand, we analyzed all possible intermediate adsorption sites, including La top (1), Pt top (2), Pt-Pt bridge (3), Pt-La bridge (4), hcp-type hollow (5), and fcc-type hollow (6) (Supplementary Fig. 21b). For comparison, Pt@HCS was simulated using the undoped Pt(111) model (Supplementary Fig. 23), including top (1), bridge (2), fcc-type hollow (3), and hcp-type hollow (4) adsorption sites. It should be noted that only those locally stable adsorption configurations were taken into account for evaluating HOR activity, and the carbon support was not included in the modeling due to its negligible effect on the electrocatalytic activities of La$_1$Pt@HCS and Pt@HCS. Such model simplifications could be favorable for specifically studying the modulation function of single-atom La on metallic Pt.

As depicted in Fig. 4a and Supplementary Fig. 24, the differential charge densities show that Pt-coordinated isolated La atom has a pronounced electron-deficient nature. In an alkaline environment, electron-deficient La can act as a Lewis acid site to attract lone pairs of electrons of Lewis base OH$^-$ (σ-donation)[40]. In the simulated 0.1 M KOH electrolyte system, a much higher OH$^-$ density is accumulated around La sites (Supplementary Fig. 25), revealing that a large number of OH$^-$ can be enriched and activated at the La site[50,51]. It is inferred that single-atom La-induced Lewis acid-base interactions can kinetically drive *H removal via combined *H and *OH, resulting in enhanced HOR activity[52].

Besides, the faster HOR kinetics over La$_1$Pt NCs can also be correlated to the change of the electronic structures incurred by the presence of La atom. The projected density of states (PDOS) of the $d$ orbital for the metal surface in La$_1$Pt NCs and Pt NCs were calculated (Fig. 4b), and the PDOS of La$_1$Pt NCs system presents a high occupancy near the Fermi level, indicating that the La$_1$ embedment increases electron transfer and conductivity, which are in good agreement with the UPS results. According to Nørskov's $d$-band center theory, when the adsorbate approaches the metal surface, its orbital electronically interfaces with the metallic orbital, resulting in energy level splitting[53]. The position of the antibonding orbitals created by energy level splitting determines the adsorption strength, and antibonding states above the Fermi level are advantageous for adsorbing OH$^-$ species in alkaline HOR. In the La$_1$Pt NCs system, the $d$-orbital of La hybridizes with the $d$-orbital of Pt or the $p$-orbital of O, which allows electron transfer to occur because of the different electronegativity, and upshifts the $d$-band center from −2.75 eV of Pt NCs to −2.14 eV of La$_1$Pt NCs (Fig. 4b). On this basis, more adsorbate antibonding states are pulled above Fermi level, corroborating a promoted interaction between adsorbates and metal atoms (Fig. 4c)[54]. Moreover, the OH binding energy (OHBE) information of La$_1$Pt@HCS and Pt/C$_{com}$ can be reflected in the corresponding cyclic voltammogram curves (Supplementary Fig. 26). The La$_1$Pt@HCS manifests lower OH$^-$ adsorption potential than Pt/C$_{com}$, suggesting that the surfaces of La$_1$Pt@HCS prefer to bind OH$^-$ compared to that of Pt/C$_{com}$.

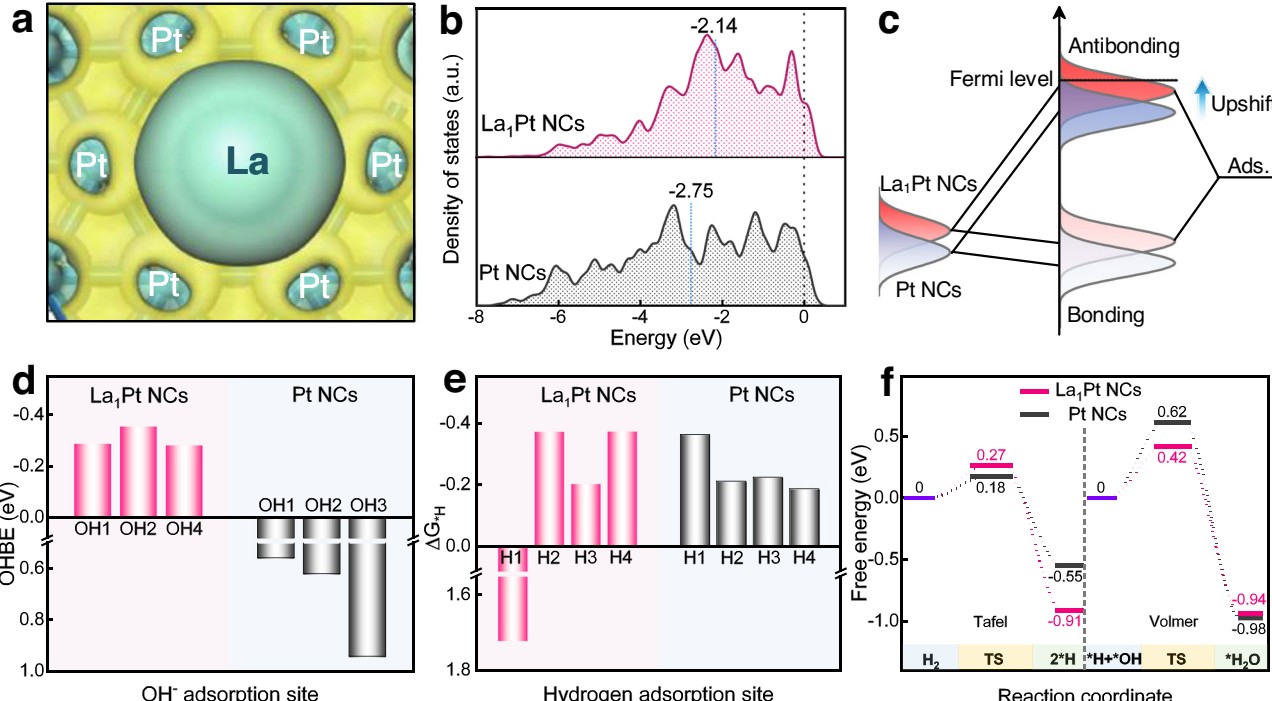

**Fig. 4 | Theoretical investigation. a** Charge density difference of La$_1$Pt NCs. Yellow, electron accumulation; cyan, electron depletion. **b** The PDOS diagram for the *d* orbital of metal surfaces on La$_1$Pt NCs and Pt NCs. The blue dotted lines are the *d*-band center. **c** Schematic DOS diagrams illustrating the single-atom La$_1$ doping on the *d*-band position of La$_1$Pt NCs. The bond formation between the model and the adsorbates (Ads.). **d** Calculated OHBE on different adsorption sites for La$_1$Pt NCs and Pt NCs. **e** $\Delta G_{*H}$ on different adsorption sites for La$_1$Pt NCs and Pt NCs. **f** Free energy diagram of the HOR (in eV) on La$_1$Pt NCs and Pt NCs.

Considering that the sluggish HOR on the Pt-based catalyst in an alkaline environment is largely related to the intrinsic poor OH⁻ adsorption on Pt (111)[29], we conducted a DFT simulation to further quantify the increased OH adsorption capacity by placing OH⁻ groups at different adsorption sites of La$_1$Pt NCs and Pt NCs. As shown in Fig. 4d, the three most stable OH⁻ adsorption sites are identified on the surface of La$_1$Pt NCs and Pt NCs upon structural optimization, and the corresponding positive OHBE values on the Pt NCs are calculated to be 0.94, 0.56, and 0.62 eV, while the OH⁻ adsorption sites on La$_1$Pt NCs deliver negative OHBE values of −0.29, −0.36, and −0.28 eV. These distinct OHBE information reveals an enhanced interaction of the OH⁻ with the surface of La$_1$Pt NCs, which is consistent with the UPS and XAS results. In addition to the OHBE, the HOR catalytic activity of La$_1$Pt NCs is also often estimated by Gibbs free energy for hydrogen adsorption ($\Delta G_{*H}$) because an optimal HOR catalyst should have $\Delta G_{*H}$ close to 0 according to the Sabatier principle[55,56]. As displayed in Fig. 4e, the La site as a Lewis acid site, has negligible adsorption capacity for H species ($\Delta G_{*H} = 1.72$ eV). Besides, the hydrogen adsorption strength of other active sites in La$_1$Pt NCs ($\Delta G_{*H} = -0.37$, −0.20, and −0.37 eV) is close to or slightly stronger than that of Pt NCs ($\Delta G_{*H} = -036$, −0.21, −0.22, and −0.19 eV), indicating that hydrogen binding energy (HBE) is not the dominant factor for boosting HOR activity in La$_1$Pt NCs. Nevertheless, while a weaker H species binding is predicted to facilitate its oxidative removal and boost HOR kinetics, slightly enhanced *H binding in La$_1$Pt NCs is essential to preserve adsorption sites from passivation by *OH.[11] It should be noted that we further modeled carbon loaded-La$_1$Pt NCs (La$_1$Pt NCs@C) and -Pt NCs (Pt NCs@C, Supplementary Fig. 27). We found that both La$_1$Pt NCs and La$_1$Pt NCs@C show similar adsorption/ desorption strength or tendency for OH (Supplementary Fig. 28a) and H intermediates (Supplementary Fig. 28b), and the same is true for Pt NCs and Pt NCs@C. These results further validate the negligible influence of carbon support on the electrocatalytic activity, highlighting the rationality of the model simplification.

We also plotted the Gibbs free energy diagram to investigate the HOR process over La$_1$Pt NCs and Pt NCs. As shown in Fig. 4f, the Tafel step happening at the optimal H adsorption site (site H3) is the first step (Supplementary Fig. 29), in which the free energies of H$_2$ dissociation for La$_1$Pt NCs and Pt NCs are determined to be 0.27 and 0.18 eV with the endothermic feature, respectively. Afterward, the reaction barriers were determined from (*H + *OH) to the water formation step in the Volmer step (Supplementary Fig. 30), which are 0.42 and 0.62 eV for the La$_1$Pt NCs and Pt NCs, respectively. The higher reaction energy barrier identifies water formation as the rate-determining step (RDS)[22], and the simulation results uncover that the reaction energy of RDS was significantly lower on La$_1$Pt NCs than on Pt NCs. We also comprehensively compared the alkaline HOR activity descriptors of the La$_1$Pt with those of other metals and metal oxides, and found that the La$_1$Pt performs best among them due to its relatively low energy barrier for the Volmer reaction as well as moderate HBE and OHBE which are not possessed by Pt(110), Ni, CeO$_2$, Pd, and PtNi alloy catalysts (Supplementary Table 5). Overall, both experimental and theoretical results jointly indicate that the enhanced alkaline HOR activity originates from the tuned structural and electronic properties of the embedded La single atom inside La$_1$Pt NCs.

As demonstrated, the weak H species adsorption capacity on rare-earth atoms cannot afford the efficient H$_2$ adsorption/dissociation kinetics in the Tafel step, and therefore introducing the rare-earth elements with high oxophilicity into metallic Pt with the optimal $\Delta G_{*H}$ could be a valid strategy to promote alkaline HOR activities, which will realize the low Volmer energy barrier by promoting the adsorption of OH⁻ species and the desorption of *H[57]. In this way, a series of Ln$_1$Pt@HCS catalysts (here Ln = Ce, Pr, Nd, and Lu) should all possess enhanced HOR catalytic activities due to the high oxophilicity of Ln metals. To verify this assumption, we performed a polarization test over all the Ln$_1$Pt@HCS electrocatalysts (Supplementary Fig. 31), and the results show that all electrocatalysts have an onset potential for

yielding response current as low as 0 V, indicating their similar energetics for alkaline HOR catalysis. Moreover, the $E_{1/2}$ values of all Ln$_1$Pt@HCS catalysts are concentrated at -10.0 mV, which is significantly lower than 38.8 mV for Pt/C$_{com}$. The increased current along with the elevated rotation rate (Supplementary Fig. 32) reveals the H$_2$ mass transport, with a linear relationship between the inverse of total current and the square root of rotational speeds (Supplementary Fig. 33). The slopes of Koutechy-Levich plots for Ce$_1$Pt@HCS, Pr$_1$Pt@HCS, Nd$_1$Pt@HCS, and Lu$_1$Pt@HCS were determined to be 5.02, 4.95, 4.49, and 4.99 cm$^2$ mA$^{-1}$ s$^{-1/2}$ at 25 mV, respectively, close to the theoretical value of 4.87 cm$^2$ mA$^{-1}$ s$^{-1/2}$. Moreover, all these Ln$_1$Pt@HCS electrocatalysts displayed remarkably enhanced CO tolerance comparing to the Pt/C$_{com}$ counterpart (Fig. 3h and Supplementary Fig. 34). DFT simulations reveal that all Ln$_1$Pt NCs (Supplementary Fig. 35) have stronger OH$^-$ adsorption capacity on Ln sites (OHBE: −0.82 to −0.38 eV) than pure Pt NCs (OHBE: 0.55 to 0.94 eV) (Supplementary Figs. 36 and 37). Notably, the correlation between the different degrees of oxophilicity for rare-earth elements and the precise catalytic activity is not the research focus of this work but will be a topic worth investigating in the future.

These results convincedly demonstrate that embedding rare-earth single atoms into Pt NCs could yield significant electronic modulation and enhance OH$^-$ adsorption strength, resulting in favorable alkaline HOR activity through accelerated Volmer kinetics. According to the electrochemical measurements and theoretical simulations, we propose the OH$^-$-induced adsorption mechanism for alkaline HOR catalysis triggered by Pt-coordinated isolated rare-earth atoms (Supplementary Fig. 38). The introduced rare-earth single-atom, as a Lewis acid site, boosts the migration of OH$^-$ inside the alkaline environment and gives rise to an increased local OH$^-$ concentration around the Pt-Ln sites, thereby improving the hydrogen oxidation ability of metallic Pt.

## Discussion

In summary, we have successfully developed a series of single-atom lanthanide (e.g., La, Ce, Pr, Nd, and Lu)-embedded Ln$_1$Pt NCs via vapor filling and spatially confined reduction of metal species inside HCS, solving the incompatibility issue between rare-earth elements and metallic Pt, and concurrently achieving high-performance HOR catalysis in alkaline media. Mechanism studies revealed that the highly oxophilic Ln$_1$ species could serve as the Lewis acid site for selective OH adsorption, and regulate the *H and *CO adsorption on Pt active sites via upshifting the $d$-band center, promoting the HOR kinetics and CO oxidation by accelerating the *H and *CO removal through OH$^-$-induced reactions. Consequently, the Ln$_1$Pt@HCS achieved remarkable electrocatalytic activity towards alkaline HOR with a mass activity of up to 14.3-times higher than that of commercial Pt/C and enhanced durability in terms of anti-CO poisoning capability and structural stability. This study exemplifies the integration of single-atom Ln$_1$ with ultrasmall Pt NCs for high-performance HOR catalysis and sheds light on the future design and fundamental understanding of metal NCs-based electrocatalysts for energy conversion.

## Methods
### Materials
Tetrapropyl orthosilicate (TPOS, 98%), ammonium hydroxide (28%–30%), resorcinol (99%), formaldehyde (37% in H$_2$O), hydrofluoric acid (40%–45%), Platinum(II) acetylacetonate (Pt(acac)$_2$, 97%), Lanthanum(III) acetylacetonate hydrate (La(acac)$_3$, 98%), Cerium(III) acetylacetonate hydrate (Ce(acac)$_3$, 99%), Praseodymium(III) acetylacetonate hydrate (Pr(acac)$_3$, 98%), Neodymium(III) 2,4-pentanedionate (Nd(acac)$_3$, 99%), Lutetium(III) 2,4-pentanedionate (Lu(acac)$_3$, 98%) were purchased from Aladdin. Carbon black (Vulcan XC-72R) was bought from Carbot Co. All reagents were used as received without further purification.

### Synthesis of HCS
Hollow carbon spheres (HCS) were prepared according to an improved method[41]. Typically, 70 mL of ethanol, 10 mL of deionized water, and 3.0 mL of ammonium hydroxide were stirred together for 10 min at room temperature, and then 3.5 mL of TPOS was added into the solution under stirring for 10 min. 2.0 mL of ethanol solution containing resorcinol (0.4 g) and formaldehyde (0.56 mL) was added to the solution. After 24 h, the SiO$_2$@resorcinol-formaldehyde resin nanospheres were collected by centrifugation, washed with ethanol three times, and then dried at 60 °C under vacuum overnight. After that, the brown powder was annealed in Ar atmosphere at 700 °C for 5 h with a heating rate of 2 °C min$^{-1}$ to produce SiO$_2$@C. The obtained black powder was etched with 1.0 M sodium hydroxide at 70 °C for 24 h. Finally, HCS material was collected by centrifugation, washed with deionized water and ethanol three times, and dried under vacuum for 12 h.

### Synthesis of La$_1$Pt@HCS
First, 0.06 mmol of Pt(acac)$_2$, 0.014 mmol of La(acac)$_3$, and 35 mg of obtained HCS were sealed under vacuum at 10$^{-5}$ mbar in quartz ampoule. Subsequently, the quartz ampoule was heated in a rotary oven at 140 °C and 200 °C for 48 h and 24 h, respectively. After the reaction, the ampoule was cooled rapidly. The powder was collected and washed with ethanol under sonication, then dried under vacuum for 12 h.

### Synthesis of Ln$_1$Pt@HCS (M = Ce, Nd, Pr, and Lu)
The synthesis of Ln$_1$Pt@HCS was the same as that of La$_1$Pt@HCS, except for the replacement of La(acac)$_3$ by Ce(acac)$_3$, Nd(acac)$_3$, Pr(acac)$_3$, and Lu(acac)$_3$ in the first step.

### Synthesis of Pt@HCS
The synthesis of Pt@HCS was the same as that of La$_1$Pt@HCS except for the absence of La(acac)$_3$.

### Synthesis of Pt/XC-72
The synthesis of Pt/XC-72 was the same as that of Pt@HCS except for the replacement of HCS by XC-72.

### Material characterizations
Scanning electron microscopy (SEM) was performed using a JSM-7900F scanning electron microscope. Transmission electron microscopy (TEM) images and low-magnification high-angle annular dark-field scanning TEM (HAADF-STEM) images were obtained with an FEL Talos F200X field emission electron microscope at an accelerating voltage of 200 kV. The signal collection duration for the HAADAF-STEM elemental maps is 10 min. Atomic resolution high magnification STEM image was obtained using FEI Themis Z aberration-corrected transmission electron microscope. Metal contents of the samples were analyzed by inductively coupled plasma optical emission spectroscopy (ICP-OES) on Thermo Scientific iCAP RRO. N$_2$ sorption analysis was performed at 77 K with a Micromeritics ASAP 2460 analyzer. X-ray powder diffraction (XRD) test was conducted on a D8 Advance X-ray powder diffractometer equipped with a Cu Kα radiation source. X-ray photoelectron spectroscopy (XPS) was performed on a K-Alpha spectrometer from Thermo Scientific using an Al Kα photon source ($hv$ = 1486.6 eV) at 12 kV. XAFS analyses were performed with Si(111) crystal monochromators at the BL14W beamline at the Shanghai Synchrotron Radiation Facility (SSRF, Shanghai, China). The XAFS spectra were recorded at room temperature using a 4-channel Silicon Drift Detector (SDD) Bruker 5040. The extended X-ray absorption fine structure (EXAFS) spectra were recorded in transmission/fluorescence mode. The Φ was extracted by ultraviolet photoelectron spectroscopy (UPS) on

ESCALAB Xi+ analyzer from Thermo Scientific. A He discharge lamp with the He (I) photo line (21.22 eV) was applied, and the high-binding energy secondary electron cutoff ($E_{cutoff}$) and the energy gap ($\Delta E$) between the valence band maximum (VBM) and the fermi level were extracted from the UPS spectra[44]. The $\Phi$ was calculated as Eq. (1):

$$\Phi = 21.22\,\text{eV} - E_{cutoff} \qquad (1)$$

The VBM position was calculated as Eq. (2):

$$E_{VBM} = 21.22\,\text{eV} - E_{cutoff} - \Delta E \qquad (2)$$

Electrochemical characterization

To prepare the electrode, 2.0 mg of catalyst, 980 µL of isopropanol, and 20 µL of 5 wt.% Nafion solution were mixed and sonicated for 1 h to prepare catalyst ink. A certain amount of the catalyst ink was taken out and coated on a glassy carbon rotating disk electrode (RDE) (diameter: 4 mm) to achieve a loading amount of 10 $\mu g_{Pt}\,cm^{-2}$. 20 wt.% Pt/C was employed as a reference (10 $\mu g_{Pt}\,cm^{-2}$). All electrochemical measurements were performed on a standard three-electrode system by a CHI 660d electrochemical workstation. All reference potentials have been converted to a reversible hydrogen electrode (vs. RHE) and were iR-corrected (i, current; R, resistance) for the uncompensated Ohmic drop. A graphite rod and a Hg/HgO electrode were employed as the counter electrode and reference electrode, respectively. Linear sweep voltammetry (LSV) was carried out in a 0.1 M of KOH solution saturated with $H_2$ at various rotation rates at a sweep speed of 5 mV s$^{-1}$. Cyclic voltammetry (CV) was performed in an Ar-saturated 0.1 M KOH solution at 50 mV s$^{-1}$. For CO-stripping voltammetry, the electrode potential was held at 0.1 V vs. RHE for 10 min to adsorb CO on the surface of Pt. The electrode was quickly moved to a fresh 0.1 M KOH solution. Then, the adsorbed CO was stripped by scanning between 0.05 and 1.25 V (vs. RHE) at a scan rate of 50 mV s$^{-1}$. Electrochemical impedance spectroscopy (EIS) was conducted with a frequency ranging from 0.1 to $10^5$ Hz and an amplitude of 5 mV under an overpotential of 30 mV[29,58].

The kinetic current density ($j_k$) can be obtained by the Koutechy–Levich (K–L) equation:

$$\frac{1}{j} = \frac{1}{j_k} + \frac{1}{j_d} \qquad (3)$$

where $j$ and $j_d$ represent the measured current and diffusional current, respectively.

$$j_d = 0.62nFD^{3/2}v^{-1/6}C_0\omega^{1/2} = BC_0\omega^{1/2} \qquad (4)$$

where $n$, $F$, $D$, $v$, $C_0$, $B$, and $\omega$ are the number of electrons transferred (2), the Faraday constant (96485 C mol$^{-1}$), the diffusion coefficient of $H_2$ ($3.7 \times 10^{-5}$ cm$^2$ s$^{-1}$), the kinematic viscosity ($1.01 \times 10^{-2}$ cm$^2$ s$^{-1}$), the solubility of $H_2$ ($7.33 \times 10^{-4}$ mol L$^{-1}$), the Levich constant, and the rotating speed, respectively.

The exchange current density ($j_0$) can be obtained by fitting $j_k$ into the Butler–Volmer (B–V) Eq. (5):

$$j_k = j_0\left(e^{\frac{\alpha F}{RT}\eta} - e^{-\frac{(1-\alpha)F}{RT}\eta}\right) \qquad (5)$$

where $\alpha$, $R$, $T$, and $\eta$ represent the transfer coefficient, the universal gas constant (8.314 J mol$^{-1}$ K$^{-1}$), the operating temperature (298 K), and the overpotential, respectively.

## Computational details and theoretical models
All calculations were carried out with the spin-unrestricted density functional theory (DFT) method by the DMol$^3$ module of the Materials

Studio software[59,60]. The exchange-correlation interactions were treated by generalized gradient approximation (GGA) with the Perdew–Burke–Ernzerhof (PBE) functional[61]. Grimme's DFT-D3 corrected method was used to account for weak interactions such as long-rage van der Waals intermolecular and dispersive intermolecular interactions[62]. The convergence criterion was set to $10^{-6}$ au in self-consistent field computation, and the real space cutoff radius of atomic orbital was set as 4.9 Å. The geometry optimization was carried out until the convergence threshold of $1 \times 10^{-5}$ Ha for energy, $2 \times 10^{-3}$ Ha Å$^{-1}$ for force, and $5 \times 10^{-3}$ Å for displacement. The Brillouin zone was sampled with the $4 \times 4 \times 1$ k-points for the geometry structures and energy calculations for the slab models[63]. The transition states were searched by using the linear synchronous transition/quadratic synchronous transit method and confirmed by the frequency calculation[64].

The optimized lattice constant of bulk face-centered cubic Pt structure is 3.866 Å, which is according well to the experimental results (3.923 Å)[65], suggesting the reliability of the chosen computational strategy. A ($4 \times 4$) Pt(111) slab model of four layers was used for Pt and La$_1$Pt, with the bottom two layers of Pt atoms fixed to mimic bulk structure. Moreover, a vacuum layer of 20 Å was set in the z-direction to avoid the interaction between the periodic layers.

The formation energies for adsorbed and embedded cases were determined by Eqs. (6) and (7):

$$E_f(Ln-adsorbed) = E\left[Ln_1Pt(111)\right] - E[Pt(111)] - \mu_{Ln}[Ln_{bulk}] \qquad (6)$$

$$E_f(Ln-embedded) = E\left[Ln_1Pt(111)\right] - E[Pt(111)] - \mu_{Ln}\left[Ln_{bulk}\right] + \mu_{Pt}\left[Pt_{bulk}\right] \qquad (7)$$

The adsorption energy of an adsorbate ($\Delta E$) was calculated by Eq. (8):

$$\Delta E = E_{absorbate/sub} - (E_{absorbate} + E_{sub}) \qquad (8)$$

where $E_{adsorbate/sub}$ is the total energy of the adsorbed system, $E_{adsorbate}$ is the energy of an adsorbate, $E_{sub}$ is the energy of the substrate.

The adsorption free energies ($\Delta G$) were determined by Eq. (9):

$$\Delta G = \Delta E + \Delta ZPE - T\Delta S \qquad (9)$$

where $\Delta E$, $\Delta ZPE$, and $\Delta S$ represent the binding energy, zero-point energy change, and entropy change of the adsorption of adsorbates, respectively.

The d-band center ($\varepsilon_d$) of the catalysts was represented as Eq. (10):

$$\varepsilon_d = \frac{\int_{-\infty}^{+\infty} E\rho_d(E)dE}{\int_{-\infty}^{+\infty} \rho_d(E)dE} \qquad (10)$$

where $\rho_d(E)$ is the density of d states at an energy level E.

## Data availability
The data supporting the conclusions of this study are present in the paper and the Supplementary Information. The raw data sets used for the presented analysis within the current study are available from the corresponding authors upon request.

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

## Acknowledgements
This work was supported by the National Natural Science Foundation of China (51877216, 52277229, 21805307, 21905300, and 22071127), National Key Research and Development of China (2022YFA1503400), the Natural Science Foundation of Shandong Province (ZR2020MB078), and Taishan Scholar Foundation (tsqn201812074).

## Author contributions
W.X. and X.Y. lead the present work. X.W. and Y.T. carried out the experiments, collected and analyzed the data. Y.T. carried out the DFT calculations. W.F., P.L., X.L., Y.C., T.C., L.Z., Q.X., and Z.Y. helped with data analysis. X.W. and X.Y. co-wrote the manuscript. All authors discussed the results and commented on the manuscript.

## Competing interests
The authors declare no competing interests.
