## [Peer Review File · Nature Communications]

REVIEWER COMMENTS

Reviewer #1 (Remarks to the Author):

In the manuscript entitled "Embedding oxophilic rare-earth single atom in platinum nanoclusters for efficient hydrogen electro-oxidation," the authors have conducted combined experimental and theoretical investigations for the hydrogen oxidation activity using Pt-based single atom catalysts (SAC). The hydrogen oxidation in alkaline medium is extremely challenging in the electrocatalysis field. This reaction is being currently the focus of intensive studies in recent years due to its importance in fuelcells. On the other hand, synthesis of SAC is one of the major challenges in reality. However, the authors have successfully synthesized multiple rare-earth based SACs in Pt nanoclusters. The manuscript may be apt for the journal. In the present form, the manuscript is missing many key information, which can be found below and must be addressed.

1. Hydrogen oxidation activity could be controlled by several descriptors such as the H-, OH- and H₂O binding energies. A comparison of the HOR activity in terms all these descriptors with other metals and metal oxides would be informative.
2. The Hydrogen oxidation activity is measured for the Ln₁Pt@HCS and/or Pt@HCS. However the modeling is done only for Pt. The effect of carbon support may have significant influence on the electrocatalytic activity. It should be described appropriately.
3. Providing the anti-CO poisoning effect for other single metal atoms in Pt would be extremely important.
4. Why embedding not depositing? Is there any experimental evidence? It should be clarified.
5. In (Figs. 21 and 22)-SI, the information about the bond lengths, bond angles for the reactant, TS and product geometries should be provided along with the change in catalysts structure as an effect of La dopants.
6. In Fig. 24, the bond lengths for the various dopants as well as the OH should be provided. Determining the formation energies for all the doped systems would be important.

7. There are some important studies regarding the hydrogen oxidation activity both theoretically and experimentally [ACS Catalysis 11 (5), 2561-2571, ACS Catalysis 12 (17), 10894-10904]. The authors should consider citing those.

8. A proper motivation for considering the Pt(111) substrate as a model for the Pt NCs should be mentioned.

Reviewer #2 (Remarks to the Author):

The authors present a new series of catalysts for hydrogen oxidation, leveraging single lanthanide atoms dispersed within ultrasmall Pt nanoparticles. These are created with a new synthesis methodology using hollow carbon spheres as reactors for confinement. The synthesis is motivated by current mechanistic understanding, which suggests that an additional catalytic site to act as a Lewis acid for specific OH binding will enhance catalytic efficiency. A thorough suite of characterization and performance measurements are included, confirming that the desired catalyst was indeed produced and that it displays enhanced efficiency and stability. These effects are explained via first principles calculations of binding energies and activation energies.

This is a very complete study showing clear improvements over existing catalysts by employing directed, hypothesis-driven design. I only have two minor comments:

1) The EXAFS characterization shown in Figure 2 provides excellent evidence that the incorporated La atoms are isolated, due to the lack of any La-La scattering. But I am not convinced by the “peak” the authors ascribe to Pt-La scattering. In the Pt EXAFS, this peak is at the same level as the noise in the Fourier-transformed data, and indeed looks very similar to another “peak” near 1 Angstrom which I’m sure the authors would agree is not related to any actual scattering. The wavelet transforms also show no indication of Pt-La scattering. In the La EXAFS, the signal for the Pt-La scattering is similarly small and on the same order as the noise, although in this case the authors do assign a peak in the wavelet transform to this scattering. Still, given the small signals, I remain skeptical. I’m aware that the EXAFS data can be fit with the single-atom La model, but I think the best evidence for how La incorporates into the structure is from the high resolution TEM imaging in Figure 1h-i.

On a related note, the La EXAFS does show clear La-O scattering, is it possible that the catalyst structure involves bridging oxygens between the Pt and La atoms, rather than alloyed La?

2) The authors conclude by emphasizing that the new catalysts outperform commercial Pt/C by a factor of 14. Given the complex synthesis procedure and rarity of lanthanides, however, I wonder if the new catalysts could be feasibly scaled in the same way as Pt/C? I know this is a fundamental study concerned mostly with controlling catalytic mechanisms, but the authors might comment on whether this strategy could be easily scaled or if there would be challenges in doing so.

A few minor tweaks as well:

Line 206-207: “extended” should be on line 207 before “EXAFS” instead of before “XANES”.

XANES in Fig 2: suggest plotting a smaller energy range so the shifts in white line energy are easier to see. The full energy range could be shown in an SI figure instead, if necessary.

Line 231-234: The authors state that the La L3-edge XANES white line intensity decrease indicates a decrease in La oxidation state, citing ref. 30. What ref. 30 actually says is that the decrease in white line intensity indicates a lower coordination number for La. A shift in oxidation state should instead show an energy shift in the white line position (similar to that between the La₂O₃ and La(OH)₃ standards, although those formally have the same La +3 oxidation state). The authors should correct this sentence. The low white line intensity for the La₁Pt@HCS catalyst is consistent with the low La coordination of atomically dispersed La within the Pt particles, but does not necessarily indicate anything about the oxidation state.

Line 481: “trigged” should be “triggered”.

Reviewer #3 (Remarks to the Author):

The catalytic effect observed on the newly developed catalysts is indeed significant, and the results worth being paid attention to. Moreover, I would personally feature the synthetic technique – I find it to be elegant. However, since the main parameter indicating the catalytic improvement is, of course, the registered current in HER and the numbers derived after normalizing the measured values, most of my questions are related to the accuracy and reproducibility:

1) How reproducible are the catalytic parameters of the catalysts from batch-to-batch? How many batches were synthesized (what is the amount of the catalyst in one batch) and how many tests were conducted per one batch? When there is a significant improvement of the catalytic activity, I think the reproducibility is a key parameter to be tested next.

2) The commercial Pt-based catalysts, used as references in the manuscript, are synthesized on the carbon black supports with particle sizes of 20-30-40 nm, I guess. The material suggested by the authors, is composed of “huge” (200 um in diameter) porous carbon spheres. And my question is how one can easily distribute those particles (2 mg of the catalyst) in the suspension and reach the formation of the stable one. To my opinion, it is challenging to suspend the material, then it is hard to apply the catalyst in the amount equal to 10 ug of Pt per 1 cm². Thus, what is the error in the catalyst loading? How can one make sure that one has not applied much more Pt on the RDE?

3) Partially answering my own question regarding the batch-to-batch reproducibility, from Figure 5a it seems that disregarding the doping element, the catalytic activity stays almost absolutely the same. Then my question is how can the materials from different experiments plus different batches plus with totally different doping elements have almost identical activity? And the other question is why are the catalytic activities of the materials with different dopants (and consequently different work functions – the authors refer to the work function change as one of the reasons of the catalytic activity improvement) are very similar?

Another comment of mine is to address the novelty of the manuscript (Introduction, page 3). I would say, that the latest literature on the development of the HOR catalysts for alkaline media (both PGM-based and Ni-based) is quite abundant and explores exactly the same three directions as the authors of this manuscript do (namely, doping of the main catalyst to obtain active sites with different functionalities, atomic level distribution of the dopant and the smallest particles possible for the main catalyst). To my opinion, the authors should have provided a more comprehensive analysis of the existing literature, in a concise form. Then, it will be easier to evaluate the novelty of this manuscript.

More technical comments:

1) What is the duration of signal collection for the HAADF-STEM element maps?

2) What are the constants values (O₂ diffusion coefficient, O₂ solubility, electrolyte viscosity) used for the calculations?

3) I was wondering why most of the catalytic parameters (page 13-14) are provided at the overpotential of 25 mV, but the comparison with the literature is done at 50 mV.

4) I wish to see the full CVs in 0.1M KOH for all the characterized materials. They don't have to be necessarily included in the SI, if they don't have special outstanding features, however from Figure 4f one can see that the newly developed catalysts do have very specific features.

5) I would omit using Figure 5 in the main part of the manuscript.

6) The CO-stripping measurements are conducted in 0.1M HClO₄. Why? And I wonder whether the Ln elements are stable in acidic media.

Response to Reviewers' Comments and Revisions Made

Response to Reviewer #1's Comments:

In the manuscript entitled "Embedding oxophilic rare-earth single atom in platinum nanoclusters for efficient hydrogen electro-oxidation," the authors have conducted combined experimental and theoretical investigations for the hydrogen oxidation activity using Pt-based single atom catalysts (SAC). The hydrogen oxidation in alkaline medium is extremely challenging in the electrocatalysis field. This reaction is being currently the focus of intensive studies in recent years due to its importance in fuel cells. On the other hand, synthesis of SAC is one of the major challenges in reality. However, the authors have successfully synthesized multiple rare-earth based SACs in Pt nanoclusters. The manuscript may be apt for the journal. In the present form, the manuscript is missing many key information, which can be found below and must be addressed.

Reply: We are happy that the reviewer acknowledges our manuscript in terms of significance, novelty, and broad interest. We especially appreciate the insightful comments/suggestions from the reviewer for further improving the quality of the manuscript.

1. Hydrogen oxidation activity could be controlled by several descriptors such as the H, OH and H₂O binding energies. A comparison of the HOR activity in terms all these descriptors with other metals and metal oxides would be informative.

Reply: Thank you for the valuable comment. According to the reviewer's good suggestion, we have compared the HOR activity of our La₁Pt with that of other metals and metal oxides in the key HOR descriptors including [Tafel reaction barrier (H₂-DE), Volmer reaction barrier (H₂O formation), HBE, OHBE, and H₂O binding energies (H₂OBE)]. As shown in Supplementary Table 5, it is found that the La₁Pt performs best among those conventional HOR catalysts due to its relatively low energy barrier for the Volmer reaction as well as moderate HBE and OHBE which are not possessed by Pt(110), Ni, CeO₂, Pd, and PtNi alloy catalysts. In this revision, we have supplemented the relevant discussion in the manuscript and the Supplementary Table 5 in the Supplementary Information.

Revision:

Paragraph 2, Page 18

"We also comprehensively compared the alkaline HOR activity descriptors of the La₁Pt with those of other metals and metal oxides, and found that the La₁Pt performs best among them due to its relatively low energy barrier for the Volmer reaction as well as moderate HBE and OHBE which are not possessed by Pt(110), Ni, CeO₂, Pd, and PtNi alloy catalysts (Supplementary Table 5)."

Supplementary References 13, 14, and 15, Page 23 in Supplementary Information

[13] García-Melchor M, López N. Homolytic products from heterolytic paths in H₂ dissociation on metal oxides: The example of CeO₂. *J. Phys. Chem. C* **118**, 10921-10926 (2014).

[14] Feng Z, *et al.* Role of hydroxyl species in hydrogen oxidation reaction: A DFT Study. *J. Phys. Chem. C* **123**, 23931-23939 (2019).

[15] Sahoo S, Dekel DR, Maric R, Alpay SP. Atomistic Insights into the hydrogen oxidation reaction of palladium-ceria bifunctional catalysts for anion-exchange membrane fuel cells. *ACS Catal.* **11**, 2561-2571 (2021).

Supplementary Table 5, Page 22 in Supplementary Information

Supplementary Table 5 Comparison between the La₁Pt and other catalysts in HOR activities including Tafel reaction barrier (H₂-DE), Volmer reaction barrier (H₂O formation), HBE, OHBE, and H₂O binding energies (H₂OBE)^a.

Systems	H ₂ -DE	H ₂ O formation	HBE	OHBE	H ₂ OBE	Reference
La ₁ Pt	0.27	0.42	-0.44	-0.35	-1.43	This work
Ni(111)	0.20	0.94	-0.72	-1.14	–	13
PtNi(111)	0.15	0.93	-0.58	-0.25	–	13
Pt(110)	0	0.41	-0.62	-3.51	-0.47	14
CeO ₂	1.82	0.69	-1.34	-1.32	-0.53	15
Pd(111)	0.03	0.98	-0.69	-3.33	-0.42	15
Pd@CeO ₂	0.05	0.48	-1.29	-3.25	-0.89	15

^a All energies are in eV units. The values are rounded up to two decimal digits.

2. *The Hydrogen oxidation activity is measured for the Ln₁Pt@HCS and/or Pt@HCS. However the modeling is done only for Pt. The effect of carbon support may have significant influence on the electrocatalytic activity. It should be described appropriately.*

Reply: We are thankful for the reviewer's constructive comments. As suggested, we further simulated carbon loaded-La₁Pt NCs (La₁Pt NCs@C) and -Pt NCs (Pt NCs@C) (Supplementary Fig. 27). It is found that both La₁Pt NCs and La₁Pt NCs@C show similar adsorption/desorption strength or tendency for OH (Supplementary Fig. 28a) and H intermediates (Supplementary Fig. 28b), and the same is true for Pt NCs and Pt NCs@C. These results validated the negligible influence of carbon support on the electrocatalytic activity, and thus the carbon support in the initial modeling was excluded in order to specifically study the modulation function of single-atom La on metallic Pt considering the same carbon support effect on La₁Pt@HCS and Pt@HCS. To clarify this issue, we have supplemented the discussion about the carbon support effects on the binding strength of the intermediates in the revised manuscript.

Revision:

Supplementary Figs. 27, 28, Pages 14-15 in Supplementary Information

Supplementary Fig. 27 Optimized models. **a**, La₁Pt NCs@C; **b**, Pt NCs@C.

Supplementary Fig. 28 Binding strength of HOR intermediates for La₁Pt and Pt with or without carbon support. **a**, OHBE; **b**, ΔG^*_{H} .

Paragraph 1, Page 16

“...the carbon support was not included in the modeling due to its negligible effect on the electrocatalytic activities of La₁Pt@HCS and Pt@HCS. Such model simplifications could be favorable for specifically studying the modulation function of single-atom La on metallic Pt.”

Paragraph 1, Page 18

“It should be noted that we further modeled carbon loaded-La₁Pt NCs (La₁Pt NCs@C) and -Pt NCs (Pt NCs@C, Supplementary Fig. 27). We found that both La₁Pt NCs and La₁Pt NCs@C show similar adsorption/desorption strength or tendency for OH (Supplementary Fig. 28a) and H intermediates (Supplementary Fig. 28b), and the same is true for Pt NCs and Pt NCs@C. These results further validate the negligible influence of carbon support on the electrocatalytic activity, highlighting the rationality of the model simplification.”

3. Providing the anti-CO poisoning effect for other single metal atoms in Pt would be extremely

important.

Reply: We are grateful for the reviewer’s valuable comments. As suggested, we have supplemented the experimental results about the anti-CO poisoning effects for other Ln₁Pt@HCS electrocatalysts measured with the same test parameters. As shown in Supplementary Fig. 34, all Ln₁Pt@HCS electrocatalysts displayed remarkably enhanced CO tolerance when comparing to Pt/C_{com} electrocatalyst (Fig. 3h), suggesting the significant role of Ln₁ dopant in promoting the CO oxidation through accelerating the combination of OH⁻ and *CO in kinetics and thermodynamics.

Revision:

Paragraph 1, Page 20

“Moreover, all these Ln₁Pt@HCS electrocatalysts displayed remarkably enhanced CO tolerance comparing to the Pt/C_{com} counterpart (Fig. 3h and Supplementary Fig. 34).”

Supplementary Fig. 34, Page 18 in Supplementary Information

Supplementary Fig. 34 CO tolerance measurements. Chronoamperometry (*j-t*) response of other Ln₁Pt@HCS catalysts in CO (1000 ppm)/H₂-saturated 0.1 M KOH at 100 mV (vs. RHE).

4. Why embedding not depositing? Is there any experimental evidence? It should be clarified.

Reply: We highly appreciate the reviewer’s keen insight in this issue. We admit that we are not confident to clearly identify whether it is “embedding” or “depositing” based on both XANES/FT-EXAFS analysis (Fig. 2h, i) and the aberration-corrected HAADF-STEM image of La₁Pt@HCS (Fig. 1h). To clarify this issue, we employed DFT simulations to analyze the formation energy (E_f) for the embedded and adsorbed cases (*ACS Catal.* 2021, 11, 2561), and the calculation formulas of E_f are as follows:

$$E_f(\text{Ln} - \text{adsorbed}) = E[\text{Ln}_1\text{Pt}(111)] - E[\text{Pt}(111)] - \mu_{\text{Ln}}[\text{Ln}_{\text{bulk}}]$$

$$E_f(\text{Ln} - \text{embedded}) = E[\text{Ln}_1\text{Pt}(111)] - E[\text{Pt}(111)] - \mu_{\text{Ln}}[\text{Ln}_{\text{bulk}}] + \mu_{\text{Pt}}[\text{Pt}_{\text{bulk}}]$$

The results reveal that the La₁-embedded configuration has lower E_f (-4.031 eV, Supplementary Fig. 21) than the La₁-deposited one (hcp site: -3.077 eV, fcc site: -3.151 eV; Supplementary Fig. 22), which provides supportive evidence for the formation of La₁-embedded La₁Pt@HCS. In this revision, we have included Supplementary Fig. 22 into the Supplementary Information, and added several lines in the manuscript to discuss it.

Revision:

Supplementary Fig. 22, Page 12 in Supplementary Information

Supplementary Fig. 22 Models of adsorbed La_1 on Pt(111). **a**, adsorbed La at the hcp site. **b**, adsorbed La at the fcc site. Unit of bond length: Å. Pt, dark blue; La, dark cyan. Bottom values represent formation energies.

Paragraph 1, Page 16

“Such a La_1 -embedded configuration has lower formation energy (-4.031 eV) than the La_1 -deposited one (fcc site: -3.151 eV, hcp site: -3.077 eV; Supplementary Fig. 22), further supporting that the La_1 was embedded in Pt NCs.”

Methods, Computational details and theoretical models, Page 23

“The formation energies for adsorbed and embedded cases were determined by Eqs. (6) and (7):

$$E_f(\text{Ln} - \text{adsorbed}) = E[\text{Ln}_1\text{Pt}(111)] - E[\text{Pt}(111)] - \mu_{\text{Ln}}[\text{Ln}_{\text{bulk}}] \quad (6)$$

$$E_f(\text{Ln} - \text{embedded}) = E[\text{Ln}_1\text{Pt}(111)] - E[\text{Pt}(111)] - \mu_{\text{Ln}}[\text{Ln}_{\text{bulk}}] + \mu_{\text{Pt}}[\text{Pt}_{\text{bulk}}] \quad (7)''$$

5. In (Figs. 21 and 22)-SI, the information about the bond lengths, bond angles for the reactant, TS and product geometries should be provided along with the change in catalysts structure as an effect of La dopants.

Reply: Thank you for the valuable comment and suggestion. We first provided the information about the bond length changes after the La doping (Supplementary Fig. 21). After that, we also refined the geometries (e.g., bond lengths and bond angles) of reactants, TS, and products for the Tafel step (Supplementary Fig. 29) and Volmer step (Supplementary Fig. 30) as suggested (Note: the mentioned Figs. 21 and 22-SI are now renumbered as Supplementary Figs. 29 and 30).

Revision:

Supplementary Fig. 21, Page 12 in Supplementary Information

Supplementary Fig. 21 Optimized La_1Pt NCs model. **a**, Stable configuration of La_1Pt . **b**, Possible adsorption sites for reaction intermediates. Unit of bond length: Å. Pt, dark blue; La, dark cyan.

Supplementary Fig. 29, Page 15 in Supplementary Information

Supplementary Fig. 29 The Tafel reaction on La_1Pt NCs models. **a**, Reactant; **b**, transition state (TS); **c**, Product. Unit of bond length: Å. Pt, dark blue; La, dark cyan; H, white.

Supplementary Fig. 30, Page 16 in Supplementary Information

Supplementary Fig. 30 The water formation reaction on La_1Pt NCs models. **a**, Reactant, **b**, TS; **c**, Product. Unit of bond length: Å. Pt, dark blue; La, dark cyan; O, red; H, white.

6. In Fig. 24, the bond lengths for the various dopants as well as the OH should be provided. Determining the formation energies for all the doped systems would be important.

Reply: We are grateful for the reviewer's good suggestions. As suggested, we have supplemented bond lengths (Supplementary Fig. 35), as well as the OH-Ln bond lengths (Supplementary Fig. 36) for all Ln_1Pt models. In addition, we also calculated the formation energies for different Ln_1Pt systems, which are shown below the corresponding models (Supplementary Fig. 35).

Revision:

Supplementary Fig. 35, Page 18 in Supplementary Information

Supplementary Fig. 35 Optimization of other Ln_1Pt NCs models. **a**, Ce_1Pt ; **b**, Pr_1Pt ; **c**, Nd_1Pt ; **d**, Lu_1Pt . Unit of bond length: Å. Pt, dark blue; Ln, another color. Bottom values represent formation energies for the Ln_1Pt system.

Supplementary Fig. 36, Page 18 in Supplementary Information

Supplementary Fig. 36 Model of OH adsorption on Ln_1 sites. **a**, Ce_1Pt ; **b**, Pr_1Pt ; **c**, Nd_1Pt ; **d**, Lu_1Pt . Unit of bond length: Å.

7. There are some important studies regarding the hydrogen oxidation activity both theoretically and experimentally [ACS Catalysis 11 (5), 2561-2571, ACS Catalysis 12 (17), 10894-10904]. The authors should consider citing those.

Reply: We greatly appreciate the reviewer for the kind reminder in citing these key references. The references provided by the reviewer have broadened our horizons on the hydrogen oxidation both theoretically and experimentally. On the basis, we have cited the references (as Refs. 49 and 57) in the manuscript for providing more inspirations for audience.

Revision:

Paragraph 1, Page 16

“Such a La_1 -embedded configuration has lower formation energy (-4.031 eV) than the La_1 -deposited one (fcc site: -3.151 eV, hcp site: -3.077 eV; Supplementary Fig. 22), further supporting that the La_1 was embedded in Pt NCs⁴⁹.”

Paragraph 1, Page 19

“As demonstrated, the weak H species adsorption capacity on rare-earth atoms cannot afford the efficient H_2 adsorption/dissociation kinetics in the Tafel step, and therefore introducing the rare earth elements with high oxophilicity into metallic Pt with the optimal ΔG_{*H} could be a valid strategy to promote alkaline HOR activities, which will realize the low Volmer energy barrier by promoting the adsorption of OH^- species and the desorption of $*\text{H}$ ⁵⁷.”

Reference 49 and 57

[49] Sahoo S, Dekel DR, Maric R, Alpay SP. Atomistic insights into the hydrogen oxidation reaction of palladium-ceria bifunctional catalysts for anion-exchange membrane fuel cells. *ACS Catal.* **11**, 2561-2571 (2021)

[57] Pagliaro MV, *et al.* Improving alkaline hydrogen oxidation activity of palladium through interactions with transition-metal oxides. *ACS Catal.* **12**, 10894-10904 (2022)

8. *A proper motivation for considering the Pt(111) substrate as a model for the Pt NCs should be mentioned.*

Reply: We are grateful for the reviewer's valuable comments. There are two motivations for considering the Pt(111) substrate as a model for the Pt NCs: (1) For fcc-Pt, the close-packed (111) surface is selected in our DFT calculations because the (111) planes possess the lowest surface energy and the most-abundant surface sites in Pt crystal (*Nat. Mater.* 2018, 17, 1033; *ACS Appl. Mater. Interfaces* 2020, 12, 40248). The Pt(111) surface was employed to model Pt NCs in DFT simulations can also be found in some excellent works, such as *J. Am. Chem. Soc.* 2023, 145, 4088; *Nat. Mater.* 2018, 17, 1033; *Nat. Catal.* 2022, 5, 923. (2) In alkaline HOR, *H₂O is generated via the slow Volmer step (as the RDS) on Pt catalysts, which is closely related to the inherently weak *OH adsorption of Pt(111) (*Science* 2011, 334, 1256; *Appl. Catal. B-Environ.* 2021, 299, 120640). As early as the 1990s, Markovic *et al.* (*J. Chem. Soc.* 1996, 92, 3719) also reported that HER and HOR on various Pt single crystals are very different, with the lowest catalytic activity of Pt(111) compared to that of other crystal planes in an alkaline environment. Thus, comparison in the binding strength of intermediates and transition state energy barrier of the Volmer reaction on the Pt(111) surface would yield more trustworthy evidence to evaluate the ease with which the reaction occurs (*J. Am. Chem. Soc.* 2020, 142, 4985). The relevant descriptions have been added to the manuscript.

Revision:

Paragraph 1, Page 15

“The Pt(111) surface is selected in DFT calculations because the (111) plane possesses the lowest surface energy and the most-abundant surface sites in Pt crystal³³.”

Response to Reviewer #2's Comments:

The authors present a new series of catalysts for hydrogen oxidation, leveraging single lanthanide atoms dispersed within ultrasmall Pt nanoparticles. These are created with a new synthesis methodology using hollow carbon spheres as reactors for confinement. The synthesis is motivated by current mechanistic understanding, which suggests that an additional catalytic site to act as a Lewis acid for specific OH binding will enhance catalytic efficiency. A thorough suite of characterization and performance measurements are included, confirming that the desired catalyst was indeed produced and that it displays enhanced efficiency and stability. These effects are explained via first principles calculations of binding energies and activation energies.

This is a very complete study showing clear improvements over existing catalysts by employing directed, hypothesis-driven design. I only have two minor comments:

Reply: We greatly appreciate the reviewer's acknowledgment in the significance and novelty of our manuscript. We also believe that this manuscript could shed light on the design of other high-

performance Pt-based nanocluster electrocatalysts and show its impact in multi-disciplinary fields.

1) The EXAFS characterization shown in Figure 2 provides excellent evidence that the incorporated La atoms are isolated, due to the lack of any La-La scattering. But I am not convinced by the “peak” the authors ascribe to Pt-La scattering. In the Pt EXAFS, this peak is at the same level as the noise in the Fourier-transformed data, and indeed looks very similar to another “peak” near 1 Angstrom which I’m sure the authors would agree is not related to any actual scattering. The wavelet transforms also show no indication of Pt-La scattering. In the La EXAFS, the signal for the Pt-La scattering is similarly small and on the same order as the noise, although in this case the authors do assign a peak in the wavelet transform to this scattering. Still, given the small signals, I remain skeptical. I’m aware that the EXAFS data can be fit with the single-atom La model, but I think the best evidence for how La incorporates into the structure is from the high resolution TEM imaging in Figure 1h-i.

On a related note, the La EXAFS does show clear La-O scattering, is it possible that the catalyst structure involves bridging oxygens between the Pt and La atoms, rather than alloyed La?

Reply: We greatly appreciate the reviewer’s significant comment and suggestion for further improving the quality of the manuscript, and totally agree with the reviewer’s viewpoints. As suggested, we have revised the relevant discussion properly. In Pt L-edge EXAFS spectra (Fig. 2f), a dominant peak with the feature of the Pt foil and a second-intensive peak similar to that of the PtO₂ are observed for La₁Pt@HCS, which could be ascribed to Pt-Pt and Pt-O contributions, respectively. In parallel, the new FT-EXAFS curve of La₁Pt@HCS (Fig. 2i) manifests two main peaks between 1.0 and 2.8 Å, corresponding to the La-O scattering similar to that of La(OH)₃ and La₂O₃. Moreover, no La-La contributing peak at around 4.0 Å is detected for La₁Pt@HCS, validating that the La species exists in an isolated monoatomic state without long-range coordination to other La centers. These findings are consistent with the observation of La single atoms embedded within Pt NCs in the AC-HAADF-STEM image (Fig. 1h). Taken together, we may conclude that the oxygen bridges the La and Pt atoms, giving rise to the single-La-atom configuration embedded in Pt NCs.

Revision:

Fig. 2, Page 11

Fig. 2 **f** FT-EXAFS curves at Pt L₃-edge, where the curve is experimental data and the circle is the best fit for La₁Pt@HCS. **g** WT-EXAFS signals at Pt L-edge for La₁Pt@HCS and the references. **i** FT-EXAFS curves at La L₃-edge, where the curve is experimental data and the circle is the best fit for La₁Pt@HCS. **j** WT-EXAFS signals at La L₃-edge for La₁Pt@HCS and the references.

Paragraph 1, Page 10

“Moreover, the Pt L-edge EXAFS spectra were obtained by Fourier transformation (FT) to elucidate bonding information (Fig. 2f). As shown, a dominant peak with the feature of the Pt foil and a second-intensive peak similar to that of the PtO₂ are observed for La₁Pt@HCS, which could be ascribed to Pt-Pt and Pt-O contributions, respectively. The FT-EXAFS curves at Pt L₃-edge were fitted in *R* space (Supplementary Fig. 13), and those fitting results are summarized in Supplementary Table 2. The lower Pt-Pt average CN (5.1) and longer Pt-Pt bond length (3.05 Å) than those of Pt foils (12, 2.77 Å) reconfirm the tendency to generate weak *d*-orbital interactions and an upshift of the *d*-band center, resulting in a promoted OH⁻ adsorption capacity.”

Paragraph 1, Page 11

“There is no La-La contributing peak at around 4.0 Å in La₁Pt@HCS, validating that the La species exists in an isolated monoatomic state without long-range coordination to other La centers. This finding is consistent with the observation of La single atoms embedded within Pt NCs in the AC-HAADF-STEM image (Fig. 1h). Similarly, WT-EXAFS analysis (Fig. 2j) at La L₃-edge manifests one intensity maximum at 6.5 Å⁻¹ related to the La-O path. Taken together, we may conclude that the oxygen bridges the La and Pt atoms, giving rise to the single-La-atom configuration.”

2) The authors conclude by emphasizing that the new catalysts outperform commercial Pt/C by a factor of 14. Given the complex synthesis procedure and rarity of lanthanides, however, I wonder if the new catalysts could be feasibly scaled in the same way as Pt/C? I know this is a fundamental study concerned mostly with controlling catalytic mechanisms, but the authors might comment on whether this strategy could be easily scaled or if there would be challenges in doing so.

Reply: Thank you for the reviewer’s perceptive question. To answer this question, we conducted two scale-up experiments: (1) We scaled up the synthetic experiment of La₁Pt@HCS for more than

28 times by increasing the dosage of HCS from 35 mg to 1000 mg in a doubled-volume ampoule with the proportional feeding of other reactants (i.e., $\text{Pt}(\text{acac})_2$ and $\text{La}(\text{acac})_3$), successfully resulting in the production of gram-scale $\text{La}_1\text{Pt}@ \text{HCS}$ (Fig. R1a). XRD (Fig. R1b), XPS (Fig. R1c, d), and LSV curves (Fig. R1e) data indicated that the gram-scale $\text{La}_1\text{Pt}@ \text{HCS}$ displays similar physicochemical properties and comparable HOR catalytic activity to conventionally obtained one, suggesting the good scalability of this synthetic approach; 2) Considering the relatively complicated fabrication of HCS that may constrain the scalable production of Ln_1Pt NCs for commercial use, we further deployed commercially available porous activated carbon (Kuraray[®] YP-50, PAC for short; BET surface area: $1568.6 \text{ m}^2 \text{ g}^{-1}$, Average pore size: 3.4 nm) to replace HCS for the scalable production of $\text{Ln}_1\text{Pt}@ \text{PAC}$. Theoretically, the abundant porous channels of PAC could play the same role as that of HCS in regulating the size of Ln_1Pt NCs in this fabrication system, and the as-obtained $\text{Ln}_1\text{Pt}@ \text{PAC}$ may retain similar physicochemical properties and HOR activity to $\text{Ln}_1\text{Pt}@ \text{HCS}$. As expected, the facile synthesis of $\text{La}_1\text{Pt}@ \text{PAC}$ could be realized in our lab without optimization of experimental parameters (Fig. R2a). While the TEM characterization of $\text{La}_1\text{Pt}@ \text{PAC}$ reveals the formation of La_1Pt NCs ($\sim 3.1 \text{ nm}$) within PAC (Fig. R2b), its electrocatalytic test discloses high HOR catalytic performance with a kinetic activity (j_k) of up to 9.5-times higher than that of commercial Pt/C (Fig. R3). It should be noted that kg-scale production of $\text{La}_1\text{Pt}@ \text{HCS}$ (or $\text{La}_1\text{Pt}@ \text{PAC}$) cannot be achieved in our lab due to the equipment limitations, which may be overcome if in a customized factory. Nevertheless, these results clearly demonstrated the high feasibility of the as-developed “vaporous metal precursors filling & spatially confined in-situ NCs formation” strategy in the scalable fabrication of Ln_1Pt NCs in porous carbon support for potential commercialization.

In this revision, we have included Fig. R1 and Fig. R2 into the Supplementary Information as Supplementary Figs. 3 and 5, respectively. In parallel, we also added several lines in the manuscript to discuss this issue accordingly.

Fig. R1 Structural and HOR performance characterization of conventional $\text{La}_1\text{Pt}@\text{HCS}$ and gram-scale $\text{La}_1\text{Pt}@\text{HCS}$ electrocatalysts. **a**, Digital photo of gram-scale $\text{La}_1\text{Pt}@\text{HCS}$ electrocatalyst. **b**, XRD. **c**, Pt $4f$ XPS spectra. **d**, La $3d$ XPS spectra. **e**, HOR polarization curves in H_2 -saturated 0.1 M KOH with a scan rate of 5 mV s^{-1} at a rotating rate of 1600 rpm.

Fig. R2 TEM images of La_1Pt NCs inside PAC support. **a**, Low magnification image. **b**, High magnification image (inset: size distribution histogram of La_1Pt NCs).

Fig. R3 HOR performance characterization of $\text{La}_1\text{Pt@PAC}$ electrocatalyst. **a**, HOR polarization curves in H_2 -saturated 0.1 M KOH with a scan rate of 5 mV s^{-1} at a rotating rate of 1600 rpm. **b**, The Koutechy-Levich plots of $\text{La}_1\text{Pt@PAC}$.

Revision:

Supplementary Fig. 3, Page 3 in Supplementary Information

Supplementary Fig. 3 Structural and HOR performance characterization of conventional $\text{La}_1\text{Pt@HCS}$ and the one by gram-scale production. **a**, Digital photo of gram-scale $\text{La}_1\text{Pt@HCS}$. **b**, XRD. **c**, Pt $4f$ XPS spectra. **d**, La $3d$ XPS spectra. **e**, HOR polarization curves in H_2 -saturated 0.1 M KOH with a scan rate of 5 mV s^{-1} at a rotating rate of 1600 rpm. The gram-scale synthesis of $\text{La}_1\text{Pt@HCS}$ was realized by increasing the dosage of HCS from 35 mg to 1000 mg in a doubled-volume ampoule with the proportional feeding of $\text{Pt}(\text{acac})_2$ and $\text{La}(\text{acac})_3$, which is the maximal production capability in our lab due to the equipment limitations.

Supplementary Fig. 5, Page 4 in Supplementary Information

Supplementary Fig. 5 TEM images of La₁Pt NCs inside PAC support. **a**, Low magnification image. **b**, High magnification image (inset: size distribution histogram of La₁Pt NCs).

Paragraph 2, Page 7

“Furthermore, large-scale fabrication of La₁Pt@HCS could be easily realized using this approach (Supplementary Fig. 3). It should be mentioned that the mesoporous channels of HCS are crucial to the formation of such ultrasmall NCs because of their effective size confinement for metal species, which is supported by two control experiments: 1) using XC-72 carbon material (Vulcan®, BET surface area: 211.5 m² g⁻¹) to replace HCS but without introducing La species, resulting in the generation of large-sized Pt nanoparticles (~12.8 nm) on the XC-72 surface (Supplementary Fig. 4); 2) using commercially available porous activated carbon (Kuraray® YP-50, PAC for short; BET surface area: 1568.6 m² g⁻¹, Average pore size: 3.4 nm) to replace HCS, leading to the successful production of La₁Pt NCs (~3.1 nm) inside PAC (Supplementary Fig. 5).”

A few minor tweaks as well:

1) Line 206-207: “extended” should be on line 207 before “EXAFS” instead of before “XANES”.

Reply: Thanks, and we have corrected it accordingly.

Revision:

Paragraph 3, Page 9

“We also assessed the effect of La₁ embedment on the atomic coordination and structural signature of La₁Pt@HCS by X-ray absorption near-edge structure (XANES) and extended X-ray absorption fine structure (EXAFS) measurements.”

2) XANES in Fig 2: suggest plotting a smaller energy range so the shifts in white line energy are easier to see. The full energy range could be shown in an SI figure instead, if necessary.

Reply: We thank the reviewer for the good suggestion, and have plotted the XANES in Fig. 2e, h with a smaller energy range as requested.

Revision:

Fig. 2, Page 11

Fig. 2e Normalized XANES at Pt L₃-edge for La₁Pt@HCS with Pt metal foil and PtO₂ as references.

Fig. 2h Normalized XANES at La L₃-edge for La₁Pt@HCS with La(OH)₃ and La₂O₃ as references.

3) Line 231-234: The authors state that the La L₃-edge XANES white line intensity decrease indicates a decrease in La oxidation state, citing ref. 30. What ref. 30 actually says is that the decrease in white line intensity indicates a lower coordination number for La. A shift in oxidation state should instead show an energy shift in the white line position (similar to that between the La₂O₃ and La(OH)₃ standards, although those formally have the same La +3 oxidation state). The authors should correct this sentence. The low white line intensity for the La₁Pt@HCS catalyst is consistent with the low La coordination of atomically dispersed La within the Pt particles, but does not necessarily indicate anything about the oxidation state.

Reply: We highly appreciate the reviewer’s valuable comment in this issue. We completely agree that the intensity of the white line for La indicates coordination number information instead of oxidation state, and thus have revised the relevant discussion in the manuscript accordingly. Actually, the white line intensities gradually decline in the order of La(OH)₃, La₂O₃, and La₁Pt@HCS, suggesting a lower La coordination number in La₁Pt@HCS, which is consistent with the coordination situation of atomically dispersed La₁ within the Pt NCs.

Revision:

Paragraph 2, Page 10

“As revealed by the La L₃-edge XANES spectra (Fig. 2h), the white line intensities gradually decline in the order of La(OH)₃, La₂O₃, and La₁Pt@HCS, suggesting a lower La coordination number in La₁Pt@HCS, which is consistent with the low coordination situation of atomically dispersed La₁ within the Pt NCs⁴⁰.”

4) Line 481: “trigged” should be “triggered”.

Reply: Sorry for the mentioned typo, and we have corrected it accordingly.

Revision:

Paragraph 2, Page 20

“According to the electrochemical measurements and theoretical simulations, we propose the OH-induced adsorption mechanism for alkaline HOR catalysis triggered by Pt-coordinated isolated rare-earth atoms.”

Response to Reviewer #3’s Comments:

The catalytic effect observed on the newly developed catalysts is indeed significant, and the results worth being paid attention to. Moreover, I would personally feature the synthetic technique – I find it to be elegant. However, since the main parameter indicating the catalytic improvement is, of course, the registered current in HER and the numbers derived after normalizing the measured values, most of my questions are related to the accuracy and reproducibility:

Reply: Thank you for the valuable yet professional comments that could help to further strengthen the manuscript. The concern raised by the reviewer and the valuable comments have been completely taken into account during the revision, and please see more details below.

1) How reproducible are the catalytic parameters of the catalysts from batch-to-batch? How many batches were synthesized (what is the amount of the catalyst in one batch) and how many tests were conducted per one batch? When there is a significant improvement of the catalytic activity, I think the reproducibility is a key parameter to be tested next.

Reply: Thank you for the reviewer’s important questions, and we also think that the reproducibility of the catalytic performance is extremely crucial. In this study, we synthesized three batches of Ln₁Pt@HCS samples. 2 mg of catalyst powders was taken from each batch to prepare the catalyst ink, and independent HOR test was performed for each catalyst ink specimen from different batches. On the basis, we confirm that the catalytic activity of the Ln₁Pt@HCS samples is highly reproducible. Actually, it is well-known that the reproducibility in the HOR performance of Ln₁Pt@HCS samples is primarily dependent on their synthetic reproducibility. Therefore, we would like to showcase the synthetic reproducibility of the La₁Pt@HCS to further clarify this issue. We first performed the synthetic experiments of all Ln₁Pt@HCS electrocatalysts on the same scale. As shown, these samples present similar Ln/Pt mass fractions (based on ICP-OES, Table R1), identical Pt valence state (Fig. R4a), typical XPS peaks of Ln species (Fig. R4b-e), and excellent electrochemical performance (Fig. R5), indicating the high reproducibility. Moreover, we have also demonstrated the gram-scale synthesis of La₁Pt@HCS (Fig. R1), which further validates the robust synthetic strategy with high reproducibility and flexible scalability. In this revision, we have updated Fig. 3d by adding statistic error bars, and added one sentence in the caption of Fig. 3d to explain the error bars.

Table R1 ICP-OES data of the synthesized Ln₁Pt@HCS electrocatalysts.

Element	La ₁ Pt@HCS	Ce ₁ Pt@HCS	Pr ₁ Pt@HCS	Nd ₁ Pt@HCS	Lu ₁ Pt@HCS
Ln (wt.%)	0.977%	0.845%	0.795%	0.941%	0.909%
Pt (wt.%)	8.598%	7.584%	8.785%	7.959%	8.091%

Fig. R4 XPS spectra comparison of Ln₁Pt@HCS. **a**, Pt 4f; **b**, Ce 3d; **c**, Pr 3d; **d**, Nd 3d; **e**, Lu 4d.

Fig. R5 Electrocatalytic performance. HOR polarization curves on La₁Pt@HCS, Ce₁Pt@HCS, Pr₁Pt@HCS, Nd₁Pt@HCS, Lu₁Pt@HCS, and Pt/C_{com} in H₂-saturated 0.1 M KOH at a rotation rate of 1600 rpm with a scan rate of 5 mV s⁻¹.

Revision:

Fig. 3d, Page 15

Fig. 3d Normalized mass activity ($j_{k,m}$) at an overpotential of 25 mV (vs. RHE) and specific activity ($j_{0,s}$). The error bars (standard deviations) in **d** are calculated based on three independent testing results.

2) The commercial Pt-based catalysts, used as references in the manuscript, are synthesized on the carbon black supports with particle sizes of 20-30-40 nm, I guess. The material suggested by the authors, is composed of “huge” (200 μm in diameter) porous carbon spheres. And my question is how one can easily distribute those particles (2 mg of the catalyst) in the suspension and reach the formation of the stable one. To my opinion, it is challenging to suspend the material, then it is hard to apply the catalyst in the amount equal to 10 μg of Pt per 1 cm^2 . Thus, what is the error in the catalyst loading? How can one make sure that one has not applied much more Pt on the RDE?

Reply: Thank you for the reviewer’s interesting questions. We are sorry for the unclear expression about the size of hollow carbon spheres (HCS) that may confuse the reviewer. In fact, the diameter of the HCS is around 200 nm (Fig. R6) instead of 200 μm that the reviewer mentioned. It is known that the suspension and dispersion of carbon materials in liquid media are closely related to their particle size, and those materials with too-large or too-small sizes are difficult to suspend and disperse because of gravity-caused settling or excessive aggregation. Fortunately, the size of the HCS is comparable to that of Vulcan[®] XC-72 carbon black used in commercial 40 wt.% Pt/C (Fig. R7) (*Int. J. Hydrogen Energ.* 2020, 45, 25672; *Bull. Korean Chem. Soc.* 2012, 33, 699), which made well suspension/dispersion possible. Indeed, the excellent compatibility for different active metals also allows HCS support material to be used in various catalytic fields, such as Pt-HCS system (15 $\mu g_{Pt} cm^{-2}$) in HER (*Adv. Mater.*, 2020, 32, 1901349), and Cu-HCS system in C-C coupling (*Appl. Catal. B-Environ.*, 2022, 306, 121111), Ru-HCS system in furfural hydrogenation (*Small*, 2022, 18, 22013). The mature synthesis technology and extensive application cases suggest that the HCS can be appointed as an excellent catalyst support material.

Fig. R6 SEM images of HCS material.

Fig. R7 TEM images of Commercial Pt/C (40 wt.%). **a**, Ref.: *Int. J. Hydrogen Energ.*, 2020, 45, 25672. **b**, Ref.: *Bull. Korean Chem. Soc.* 2012, 33, 699.

Comparing to XC-72 particle carbon supports, the HCS possesses unique hollow structure, low density, uniform size, and ultra-thin carbon shell properties, which further facilitates the suspension/dispersion of HCS and the subsequent formation of a stable catalyst ink. As shown in Fig. R8, both the as-prepared La₁Pt@HCS (5 mL) and the commercial 20 wt.% Pt/C catalyst inks still maintain excellent suspension without obvious sedimentation after standing for 5 days.

Fig. R8 Digital photographs of commercial 20 wt.% Pt/C and La₁Pt catalyst inks in isopropanol. **a**, Fresh catalyst inks. **b**, Catalyst inks after standing for 5 days.

Regarding the concern about the accuracy of catalyst loading, we prepared three different batches of La₁Pt@HCS samples, and the catalyst ink prepared from each sample batch was sampled three times for error determination. To reduce weighing errors, we increased the ink sampling volume by 100 times compared to the actual coating volume on the RDE (Note: the ink sampling volume here does not represent the actual coating amount on the RDE). The catalyst ink was dropped on clean copper foil, and the actual mass of the dried catalyst was obtained by weighing after natural drying. The average sample mass of three batches was determined to be 1.46, 1.51, and 1.49 mg, and the standard deviation is only 0.02. The highly consistent catalyst extraction is attributed to the uniform and moderate-size structure of HCS. The stable and low Pt loading (8.6 wt.%) from batch-to-batch further ensures precisely controllable Pt loading on the RDE. In addition, the almost same diffusion-controlled limiting current density in the LSV curves (Supplementary Fig. 31) also supports the successful realization of the same Pt metal loading on the RDE. In summary, it can be inferred that the Pt loading on the RDE is precise and controllable during the HOR test.

3) Partially answering my own question regarding the batch-to-batch reproducibility, from Figure 5a it seems that disregarding the doping element, the catalytic activity stays almost absolutely the same. Then my question is how can the materials from different experiments plus different batches plus with totally different doping elements have almost identical activity? And the other question is why are the catalytic activities of the materials with different dopants (and consequently different work functions – the authors refer to the work function change as one of the reasons of the catalytic activity improvement) are very similar?

Another comment of mine is to address the novelty of the manuscript (Introduction, page 3). I would say, that the latest literature on the development of the HOR catalysts for alkaline media (both PGM-based and Ni-based) is quite abundant and explores exactly the same three directions as the authors of this manuscript do (namely, doping of the main catalyst to obtain active sites with different functionalities, atomic level distribution of the dopant and the smallest particles possible for the main catalyst). To my opinion, the authors should have provided a more comprehensive analysis of the existing literature, in a concise form. Then, it will be easier to evaluate the novelty of this manuscript.

Reply: We highly appreciate the reviewer for the enlightening comments. The similar limiting current density (j_l) and slope in LSV curves for different $\text{Ln}_1\text{Pt}@HCS$ electrocatalysts may be due to the following reasons: (1) It is known that the catalyst loading is related to its catalytic performance, especially for j_l . Both too high and too low Pt loadings do affect the j_l value (*ChemElectroChem*, 2020, 7, 1107). Here, the HCS support material with the unique hollow structure, low density, uniform size, and thin carbon shell yields homogeneous catalyst inks applied on the RDE. Besides, the batch-to-batch compositional stability and low Pt loading (8.6 wt.%) further enable precise and controllable Pt loading during each HOR test, resulting in the identical j_l . (2) The apparent current density (j) in the LSV curve is a combination of the kinetic current density (j_k) and diffusion current density (j_d), which can be described by the Koutecky-Levich (K-L) equation. Benefiting from the Lewis acid-base effect induced by single Ln atoms, all $\text{Ln}_1\text{Pt}@HCS$ catalysts show ultrahigh j_k (about 14 times that of commercial Pt/C). And j_d is affected by the H_2 mass transfer, which is constant at the same RDE rotation speed and overpotential. Thus, in the mixed control region, a higher j_k yields a j closer to j_d , which also means that ultrahigh intrinsic j_k of $\text{Ln}_1\text{Pt}@HCS$ catalysts will then deliver similar LSV performance at the same overpotential (*J. Electrochem. Soc.* 2010, 157, B1529).

In addition, the similar catalytic activity for different Ln-doped Pt catalysts may originate from three other critical aspects besides the work function: (1) Similar intrinsic OHBE. We have proven that the improved alkaline HOR performance is largely attributable to the enhanced OH-binding strength. Here the introduced rare-earth single-atom, as a Lewis acid site, boosts the migration of OH^- inside the alkaline environment and gives rise to an increased local OH^- concentration around the Pt-Ln sites, thereby improving the hydrogen oxidation ability of metallic Pt. As shown in Fig. R9, the different Ln sites in $\text{Ln}_1\text{Pt}@HCS$ yield very similar OHBEs that are stronger than Pt NCs, supporting similar and higher alkaline HOR activity than Pt NCs. (2) Similar and stronger oxophilicity of different Ln elements. Intrinsic oxophilicity of catalysts is crucial for electrocatalytic reactions in alkaline media, which can affect the selective binding of oxygen-containing species to metal active sites (*Nat. Catal.* 2021, 4, 711). Lanthanide metals are the most oxygen-philic elements due to their very low electronegativity (*Chem. Rev.* 2022, 122, 5519). Specifically, the oxophilic

strengths of La, Ce, Pr, and Nd are 1.0, 0.9, 1.0, and 1.0 (*Inorg. Chem.* 2016, 55, 9461), respectively, which are much higher than those of Pt (0.1). These almost identical oxophilic strengths for different Ln elements strongly endow them with similar alkaline HOR catalytic activities. (3) Limitations of H₂ mass transfer for these catalysts with ultra-high intrinsic activity (*J. Electrochem. Soc.* 2010, 157, B1529). In the mixed control region, a higher j_k yields a j closer to j_d , which also means that ultrahigh intrinsic j_k of Ln₁Pt@HCS catalysts will then deliver similar LSV performance at the same overpotential.

Fig. R9 OHBE. Calculated OHBE for La₁Pt NCs, Ce₁Pt NCs, Pr₁Pt NCs, Nd₁Pt NCs, Lu₁Pt NCs, and Pt NCs.

According to the reviewer's important suggestion, we have revised the introduction part in order to provide the audience with a broader yet more comprehensive landscape of the development for alkaline HOR catalysts. In other words, besides discussing Pt-based HOR catalysts, we tried our best to include the research background of other HOR catalysts (e.g., PGM-based and Ni-based) into the Introduction part, which may further improve the novelty of the manuscript and attract broader interests from diversified audience. By surveying the latest literatures about alkaline HOR catalysis, we proposed that an ideal alkaline HOR electrocatalyst should feature the following attributes: 1) in addition to the main active sites, another type of active site with selective adsorption-desorption behavior for specific reactant species should be deployed on the electrocatalyst surface to essentially avoid the mutual adsorption-desorption competition between reactant species; 2) the dissimilar active sites should be well distributed in the electrocatalyst at the atomic level and have synergistic effects with the main active sites to promote the interfacial HOR kinetics and resist CO poisoning via regulating the *d*-band centers, and concurrently strengthen the structural stability of the electrocatalyst; 3) the electrocatalyst should have an ultrasmall size to maximumly uplift its atomic utilization. In this revision, we have supplemented this section and cited relevant references in the manuscript.

Revision:

Paragraph 2, Pages 3, 4

“To radically address such issues, an ideal alkaline HOR electrocatalyst (e.g., platinum group metal (PGM)-based and Ni-based) should feature the following attributes: 1) in addition to the main active sites, another type of active site with selective adsorption-desorption behavior for specific reactant species should be deployed on the electrocatalyst surface to essentially avoid the mutual adsorption-desorption competition between reactant species^{5,24-26}; 2) the dissimilar active sites should be well distributed in the electrocatalyst at the atomic level and have synergistic effects with the main active

sites to promote the interfacial HOR kinetics and resist CO poisoning via regulating the *d*-band centers, and concurrently strengthen the structural stability of the electrocatalyst^{7,8,17,27,28}; 3) the electrocatalyst should have an ultrasmall size to maximumly uplift its atomic utilization²⁹⁻³².”

References

- [5] Zhan C, *et al.* Subnanometer high-entropy alloy nanowires enable remarkable hydrogen oxidation catalysis. *Nat. Commun.* **12**, 6261 (2021).
- [7] Ma M, *et al.* Single-atom molybdenum engineered platinum nanocatalyst for boosted alkaline hydrogen oxidation. *Adv. Energy Mater.* **12**, 2103336 (2022).
- [8] Zhang Y, *et al.* Atomically isolated Rh sites within highly branched Rh₂Sb nanostructures enhance bifunctional hydrogen electrocatalysis. *Adv. Mater.* **33**, 2105049 (2021).
- [17] Li L, *et al.* Surface and lattice engineered ruthenium superstructures towards high-performance bifunctional hydrogen catalysis. *Energ. Environ. Sci.* **16**, 156-166 (2023)
- [24] Qin S, *et al.* Ternary nickel-tungsten-copper alloy rivals platinum for catalyzing alkaline hydrogen oxidation. *Nat. Commun.* **12**, 2686 (2021).
- [25] Wang K, *et al.* The exclusive surface and electronic effects of Ni on promoting the activity of Pt towards alkaline hydrogen oxidation. *Nano Res.* **15**, 5865-5872 (2022).
- [26] Luo H, *et al.* Amorphous MoO_x with high oxophilicity interfaced with PtMo alloy nanoparticles boosts anti-CO hydrogen electrocatalysis. *Adv. Mater.* 2211854 (2023).
- [27] Mao J, *et al.* Isolated Ni atoms dispersed on Ru nanosheets: High-performance electrocatalysts toward hydrogen oxidation reaction. *Nano Lett.* **20**, 3442-3448 (2020).
- [28] Shan J, *et al.* Integrating interactive noble metal single-atom catalysts into transition metal oxide lattices. *J. Am. Chem. Soc.* **144**, 23214-23222 (2022).
- [29] Wang X, *et al.* Atomic-precision Pt₆ nanoclusters for enhanced hydrogen electro-oxidation. *Nat. Commun.* **13**, 1596 (2022).
- [30] Zhao T, *et al.* Pseudo-Pt monolayer for robust hydrogen oxidation. *J. Am. Chem. Soc.* **145**, 4088-4097 (2023).
- [31] Liu D-Q, *et al.* Tailoring interfacial charge transfer of epitaxially grown Ir clusters for boosting hydrogen oxidation reaction. *Adv. Energy Mater.* **13**, 2202913 (2023).
- [32] Yang X, *et al.* CO-tolerant PEMFC anodes enabled by synergistic catalysis between iridium single-atom sites and nanoparticles. *Angew. Chem. Int. Ed.* **60**, 26177-26183 (2021).

More technical comments:

1) *What is the duration of signal collection for the HAADF-STEM element maps?*

Reply: Thank you for the reviewer’s good question. In this study, we set the signal collection duration of HAADF-STEM elemental maps to 10 min to avoid possible agglomeration of nanoclusters due to prolonged electron beam bombardment while preserve the accuracy of data acquisition. We have supplemented the corresponding description in the manuscript.

Revision:

Methods, Material characterizations, Page 22

“The signal collection duration for the HAADF-STEM elemental maps is 10 min.”

2) *What are the constants values (O₂ diffusion coefficient, O₂ solubility, electrolyte viscosity) used for the calculations?*

Reply: Many thanks for the valuable question. In Eq. (4), n , F , D , v , C_0 , B , and ω are the number of electrons transferred (2), the Faraday constant (96485 C mol^{-1}), the diffusion coefficient of H_2 ($3.7 \times 10^{-5} \text{ cm}^2 \text{ s}^{-1}$), the kinematic viscosity ($1.01 \times 10^{-2} \text{ cm}^2 \text{ s}^{-1}$), the solubility of H_2 ($7.33 \times 10^{-4} \text{ mol L}^{-1}$), the Levich constant, and the rotating speed, respectively. In Eq. (5), α , R , T , and η represent the transfer coefficient, the universal gas constant ($8.314 \text{ J mol}^{-1} \text{ K}^{-1}$), the operating temperature (298 K), and the overpotential, respectively. We have added such information in the manuscript during the revision.

Revision:

Methods, Electrochemical characterization, Page 23

“where n , F , D , v , C_0 , B , and ω are the number of electrons transferred (2), the Faraday constant (96485 C mol^{-1}), the diffusion coefficient of H_2 ($3.7 \times 10^{-5} \text{ cm}^2 \text{ s}^{-1}$), the kinematic viscosity ($1.01 \times 10^{-2} \text{ cm}^2 \text{ s}^{-1}$), the solubility of H_2 ($7.33 \times 10^{-4} \text{ mol L}^{-1}$), the Levich constant, and the rotating speed, respectively.”

“where α , R , T , and η represent the transfer coefficient, the universal gas constant ($8.314 \text{ J mol}^{-1} \text{ K}^{-1}$), the operating temperature (298 K), and the overpotential, respectively.”

3) I was wondering why most of the catalytic parameters (page 13-14) are provided at the overpotential of 25 mV, but the comparison with the literature is done at 50 mV.

Reply: We are sorry for the confusion caused. As suggested, we have reselected some recent works that performed electrochemical tests with an overpotential of 25 mV, and then summarized and compared them in Fig. 3e. The results highlight the superior catalytic performance of $\text{La}_1\text{Pt@HCS}$ in comparison with most PGM-based electrocatalysts.

Revision:

Fig. 3e, Page 15

Fig. 3e Comparison of the $j_{k,m}$ of $\text{La}_1\text{Pt@HCS}$ at 25 mV (vs. RHE) with that of other recently reported PGM-based electrocatalysts.

Paragraph 1, Page 13

“To the best of our knowledge, the remarkable mass activity (at 25 mV) of $\text{La}_1\text{Pt@HCS}$ outperforms that of most PGM-based electrocatalysts (Fig. 3e).”

4) I wish to see the full CVs in 0.1M KOH for all the characterized materials. They don't have to be necessarily included in the SI, if they don't have special outstanding features, however from Figure 4f one can see that the newly developed catalysts do have very specific features.

Reply: We highly appreciate the reviewer's insightful comments. The full CV of La₁Pt@HCS after potential recalibration is shown in Fig. R10. As suggested, we also provided the full CV curves for all Ln₁Pt@HCS electrocatalysts (Fig. R11). While the HCS support material with a high specific surface area and mesoporous structure displays obvious electrical double-layer interference on CV curves, some clues indeed can be found from the slight peak shift. For instance, they manifest lower OH⁻ adsorption potential than Pt/C_{com}, suggesting that the surfaces of Ln₁Pt@HCS prefer to bind OH⁻ compared to that of Pt/C_{com}. In this revision, we have included Fig. R10 into the Supplementary Information as Supplementary Fig. 26, and added several sentences in the manuscript to discuss this Figure.

Fig. R10 CV curves of La₁Pt@HCS and Pt/C_{com} in Ar-saturated 0.1 M KOH solution.

Fig. R11 CV curves of Ln₁Pt@HCS in Ar-saturated 0.1 M KOH solution.

Revision:

Supplementary Fig. 26, Page 14

Supplementary Fig. 26 CV curves of La₁Pt@HCS and Pt/C_{com} in Ar-saturated 0.1 M KOH solution.

Paragraph 1, Page 17

“Moreover, the OH binding energy (OHBE) information of La₁Pt@HCS and Pt/C_{com} can be reflected in the corresponding cyclic voltammogram curves (Supplementary Fig. 26). The La₁Pt@HCS manifests lower OH⁻ adsorption potential than Pt/C_{com}, suggesting that the surfaces of La₁Pt@HCS prefer to bind OH⁻ compared to that of Pt/C_{com}.”

5) I would omit using Figure 5 in the main part of the manuscript.

Reply: Thank you for the valuable suggestion, and we have split and moved Fig. 5 to the Supplementary Information section as Supplementary Figs. 31, 33, 37, and 38 accordingly.

Revision:

Supplementary Figs. 31, 33, 37, and 38, Pages 16, 17, and 19

Supplementary Fig. 31 Electrocatalytic performance. HOR polarization curves on La₁Pt@HCS, Ce₁Pt@HCS, Pr₁Pt@HCS, Nd₁Pt@HCS, Lu₁Pt@HCS, and Pt/C_{com} in H₂-saturated 0.1 M KOH at a rotation rate of 1600 rpm with a scan rate of 5 mV s⁻¹.

Supplementary Fig. 33 Electrocatalytic performance. The Koutechy-Levich plots of Ce₁Pt@HCS, Pr₁Pt@HCS, Nd₁Pt@HCS, and Lu₁Pt@HCS.

Supplementary Fig. 37 OHBE. Calculated OHBE on rare-earth sites and Pt sites for Ce₁Pt NCs, Pr₁Pt NCs, Nd₁Pt NCs, and Lu₁Pt NCs.

Supplementary Fig. 38 Mechanism model. Schematic illustration of the proposed HOR mechanism on Ln₁Pt NCs catalysts. Pt, dark blue. Rare-earth atom, golden. O, red. H, white.

6) The CO-stripping measurements are conducted in 0.1M HClO₄. Why? And I wonder whether the Ln elements are stable in acidic media.

Reply: Thank you for the insightful question. We are sorry for ignoring the stability issue of Ln species in acidic media. Many reports confirmed that CO stripping measurements for ECSA evaluation of Pt-based electrocatalysts can be performed in 0.1 M KOH media, such as *J. Am. Chem. Soc.* 2023, 145, 4088; *J. Am. Chem. Soc.* 2017, 139, 5156; *Angew. Chem. Int. Ed.* 2017, 56, 15594; *ACS Catal.* 2022, 12, 10894. We thus re-performed the CO stripping measurements using an alkaline media (0.1 M KOH) and revised the manuscript accordingly.

Revision:

Supplementary Fig. 17, Page 10

Supplementary Fig. 17 CO-stripping curves in 0.1 M KOH solution at a scan rate of 50 mV s⁻¹. **a**, La₁Pt@HCS; **b**, Pt@HCS; **c**, Pt/C_{com}; **d**, Pt/XC-72.

Paragraph 2, Page 13

“As the ECSA is determined to be 129.5 m² g⁻¹ for La₁Pt@HCS, 112.3 m² g⁻¹ for Pt@HCS, 42.8 m² g⁻¹ for Pt/C_{com} and 55.1 m² g⁻¹ for Pt/XC-72 based on CO-stripping measurements (Supplementary Fig. 17), the $j_{0,s}$ of La₁Pt@HCS was thus calculated to be 1.55 mA cm⁻², which is 6.7, 4.3, and 6.2 times higher than that of Pt@HCS (0.23 mA cm⁻²), Pt/C_{com} (0.36 mA cm⁻²), and Pt/XC-72 (0.25 mA cm⁻²), respectively.”

REVIEWER COMMENTS

Reviewer #1 (Remarks to the Author):

The authors have addressed all the remarks satisfactorily. The manuscript may be appropriate for publication in Nature Communications.

Reviewer #2 (Remarks to the Author):

The authors have responded the majority of my concerns. The only issue that remains is the question of bridging oxygens between Pt and La. I suggested this possibility because the EXAFS data shows clear Pt-O and La-O scattering, but no obvious La-La or Pt-La scattering. That made me wonder if the structure is actually Pt-O-La-O, instead of Pt-La-O.

I may not have been clear enough in my original review on this point, because the authors agreed that this might be the case in their response letter (and made edits to the manuscript about bridging oxygens), but the modeling in the manuscript still treats the catalyst structure as if Pt and La are directly adjacent (see e.g. Figure 4a, Supplementary Figure 27, 29, 30, and 38, etc.).

It's possible that the HAADF-STEM measurements shown in Figure 1 h,i discount the possibility of bridging oxygen atoms. Are these measurements at sufficient resolution to observe an oxygen atom between Pt and La atoms? Because there is no obvious oxygen seen in the figure to my eye.

I suspect that the authors' original interpretation of the structure (perhaps best shown in Supplementary Figure 38) may be correct, but the Pt-La scattering is not strong enough to be seen in the EXAFS measurements. If this is the case, the manuscript can move forward with publication, with a revised discussion of the EXAFS results.

But, if the authors believe that bridging oxygens between Pt and La may actually be present, the modeling would need to be redone to take this into account.

This is the only remaining issue with the manuscript from my viewpoint, and I would be happy to recommend publication once it is addressed.

Reviewer #4 (Remarks to the Author):

I am happy with the authors' detailed responses to the queries. Just two minor comments.

1. In Figure 3d, only one section has error bars (RSD), why did the other section ($j_{o,s}$) not given?

2. I saw a recent work on Pd-CeO₂ that showed an excellent performance towards HOR in both half and full cell configurations in alkaline (i.e., Ogada et al., ACS Catal. 2022, 12, 7014–7029) with values comparable to the findings but was not cited here. Do the authors have any reasons for omitting the citation of this work, and comparing the data as others in Figure 3.

Response to Reviewers' Comments and Revisions Made

Response to Reviewer #1's Comments:

The authors have addressed all the remarks satisfactorily. The manuscript may be appropriate for publication in Nature Communications.

Reply: We appreciate the reviewer's positive comments as well as the approval for the acceptance of our manuscript.

Response to Reviewer #2's Comments:

The authors have responded the majority of my concerns. The only issue that remains is the question of bridging oxygens between Pt and La. I suggested this possibility because the EXAFS data shows clear Pt-O and La-O scattering, but no obvious La-La or Pt-La scattering. That made me wonder if the structure is actually Pt-O-La-O, instead of Pt-La-O.

I may not have been clear enough in my original review on this point, because the authors agreed that this might be the case in their response letter (and made edits to the manuscript about bridging oxygens), but the modeling in the manuscript still treats the catalyst structure as if Pt and La are directly adjacent (see e.g. Figure 4a, Supplementary Figure 27, 29, 30, and 38, etc.).

It's possible that the HAADF-STEM measurements shown in Figure 1 h,i discount the possibility of bridging oxygen atoms. Are these measurements at sufficient resolution to observe an oxygen atom between Pt and La atoms? Because there is no obvious oxygen seen in the figure to my eye.

I suspect that the authors' original interpretation of the structure (perhaps best shown in Supplementary Figure 38) may be correct, but the Pt-La scattering is not strong enough to be seen in the EXAFS measurements. If this is the case, the manuscript can move forward with publication, with a revised discussion of the EXAFS results.

But, if the authors believe that bridging oxygens between Pt and La may actually be present, the modeling would need to be redone to take this into account.

This is the only remaining issue with the manuscript from my viewpoint, and I would be happy to recommend publication once it is addressed.

Reply: Thank you for the valuable yet professional comments that could help to further strengthen the manuscript. Indeed, atomic-resolution HAADF-STEM image (Fig. 1h) clearly shows that isolated single-La-atom intercalates into the Pt nanocluster without obvious bridging oxygen. This typical colloidal structure shares characteristics to the reported Au₁Pd catalysts (Nat. Mater., 2012, 11, 49-52). It gives crucial advice for constructing a Pt-La configuration to model La₁Pt catalyst with DFT simulations.

Previously, the EXAFS data delivered clear La-O scattering, but no obvious Pt-La scattering. The Pt-La scattering may not be strong enough to be seen in the EXAFS measurements. However, in the WT-EXAFS result, it is probably reasonable to assign the second-intensive peak at ~4.5 Å to the Pt-La path. Regarding the observed La-O scattering, we proposed two possible bonding mechanisms: (i) For Pt-La coordinated single atom alloy, the incorporation of O might be due to the oxidation of the La₁Pt catalyst during ex situ tests (Such as a clear Pd-O scattering in Pd₁Cu case (Nat. Nanotechnol., 2021, 16, 1386)). This phenomenon has been recorded in other single-atom alloy systems as well. For example, Ni₁Pt alloy features obvious Ni-O scattering but no Pt-Ni

scattering. Li et al. (Nat. Catal., 2019, 2, 495) suggested that the surface oxidation of Ni embedded in Pt nanocluster occurred, and thus they constructed an O-free Pt-Ni model using DFT simulations for latter analysis and discussion. (ii) The La-O scattering may belong to the Pt-O-La configuration as previously proposed by the reviewer. To verify this, DFT simulations were applied to check whether the Pt-O-La configuration features the reasonable OHBEs that well-match the excellent alkaline HOR activity. We first considered the possible incorporation sites for O atoms, including Pt-O_{hcp}-La (Fig. R1a), Pt-O_{fcc}-La (Fig. R1b), and Pt-O_{bridge}-La. DFT simulations indicated that the O at the bridge site (i.e., Pt-O_{bridge}-La) cannot form a stable configuration with Pt and would be transferred to the fcc site. Pt-O_{fcc}-La, in contrast, has a lower formation energy (-1.566 eV) than Pt-O_{hcp}-La (-1.380 eV), indicating that the O bridges Pt and La at the fcc hollow site. Based on this finding, we constructed the complete Pt-O_{fcc}-La model (Fig. R2), and further calculated OHBE at the La adsorption site (Fig. R3a, note that the OH species cannot be stably adsorbed on the coordinated Pt sites and transferred to the La sites). The Pt-La configuration shows the most favorable OHBE over Pt-O_{fcc}-La at each adsorption site for alkaline HOR catalysis (Fig. R3b). According to our electrochemical test results (Fig. 3), the alkaline HOR catalytic activity of La₁Pt@HCS is 14.3 times higher than that of Pt/C_{com}. It is clear that the Pt-O_{fcc}-La configuration cannot support the greatly enhanced alkaline HOR activity due to the undesirable OHBE. Hence, we conclude that the La-Pt bond is the main active site, while Pt-O-La bond does not contribute to the activity enhancement. In light of these findings, we employed the Pt-La configuration to model the La₁Pt electrocatalyst in DFT simulations (Supplementary Fig. 21).

Fig. R1 Models of Pt-O-La. **a**, O at the hcp site (Pt-O_{hcp}-La). **b**, O at the fcc site (Pt-O_{fcc}-La). Unit of bond length: Å. Pt, dark blue; La, dark cyan; O, red. Bottom values represent formation energies (Unit: eV).

Fig. R2 Model of complete Pt-O_{fcc}-La. Unit of bond length: Å. Pt, dark blue; La, dark cyan; O, red.

Fig. R3 OH adsorption. (a) Models of OH adsorption on La site for Pt-O_{fcc}-La. Pt, dark blue; La, dark cyan; O, red. (b) OHBE on different sites for Pt-La and Pt-O_{fcc}-La models. The details of the Pt-La model are shown in Supplementary Fig. 21

Revision:

Paragraph 2, Page 10

“The La-O bond might be due to the oxidation of the single atom alloy catalyst during ex situ tests^{13,34}.”

Paragraph 1, Page 11

“WT-EXAFS analysis (Fig. 2j) at La L₃-edge manifests one intensity maximum at ~2.5 Å related to the La-O path. It is probably reasonable to assign the second intensity peak at ~4.5 Å in the WT-EXAFS analysis to the Pt-La path. Taken together, we may conclude that single La atom is coordinated to a Pt atom, giving rise to the Pt-La configuration.”

Response to Reviewer #3’s Comments:

I am happy with the authors’ detailed responses to the queries. Just two minor comments.

Reply: We greatly appreciate the reviewer’s acknowledgment for our revised manuscript, and would be happy to further revise this manuscript according to the reviewer’s supplementary comments.

1. In Figure 3d, only one section has error bars (RSD), why did the other section ($j_{0,s}$) not given?

Reply: Thank you for the valuable comment. In this revision, we have updated Fig. 3d by adding statistic error bars for $j_{0,s}$, and added one sentence in the caption of Fig. 3d to explain the error bars.

Revision:

Fig. 3d, Page 15

Fig. 3d Normalized mass activity ($j_{k,m}$) at an overpotential of 25 mV (vs. RHE) and specific activity ($j_{0,s}$). The error bars (standard deviations) in **d** are calculated based on three independent testing results.

2. I saw a recent work on Pd-CeO₂ that showed an excellent performance towards HOR in both half and full cell configurations in alkaline (i.e., Ogada *et al.*, *ACS Catal.* 2022, 12, 7014-7029) with values comparable to the findings but was not cited here. Do the authors have any reasons for omitting the citation of this work, and comparing the data as others in Figure 3.

Reply: We greatly appreciate the reviewer for the kind reminder in citing this key reference. The reference provided by the reviewer has broadened our horizons both theoretically and experimentally on rare earth (oxide)/PGM-based HOR electrocatalysts. On the basis, we have cited this reference as Ref. 25 and Supplementary Ref. 9 (in Supplementary Table 4) for comparing catalytic performance.

Revision:

Paragraph 2, Page 3

“...in addition to the main active sites, another type of active site with selective adsorption-desorption behavior for specific reactant species should be deployed on the electrocatalyst surface to essentially avoid the mutual adsorption-desorption competition between reactant species^{5,24-26;}”

Reference 25

[25] Ogada JJ, *et al.* CeO₂ modulates the electronic states of a palladium onion-like carbon interface into a highly active and durable electrocatalyst for hydrogen oxidation in anion-exchange-membrane fuel cells. *ACS Catal.* **12**, 7014-7029 (2022).

Supplementary Reference 9, Page 23 in Supplementary Information

[9] Ogada JJ, *et al.* CeO₂ modulates the electronic states of a palladium onion-like carbon interface into a highly active and durable electrocatalyst for hydrogen oxidation in anion-exchange-membrane fuel cells. *ACS Catal.* **12**, 7014-7029 (2022).

Supplementary Table 4 (only modified part), Page 21 in Supplementary Information

Supplementary Table 4 Benchmark HOR activities and the relevant parameters of the PGM-based catalysts in alkaline electrolytes.

Catalysts	Catalyst loading [$\mu\text{g}_{\text{PGM}} \text{cm}_{\text{disk}}^{-2}$]	ECSA [$\text{m}^2 \text{g}_{\text{PGM}}^{-1}$]	$j_{0,s}$ [$\text{mA cm}_{\text{PGM}}^{-2}$]	Reference
... ..				
Pd/CB		3.2	0.038	
Pd-CeO ₂ /CB	14.4	8.8	0.136	9
Pd/OLC		7.8	0.043	
Pd-CeO ₂ /OLC		17.1	0.569	
... ..				

REVIEWERS' COMMENTS

Reviewer #2 (Remarks to the Author):

The authors have addressed all of my remaining concerns. I am happy to recommend publication.